# Relating by Contrasting: A Data-efficient Framework for Multimodal DGMs

**Yuge Shi[1], Brooks Paige[2,4], Philip H.S. Torr[1] & N. Siddharth**[*1,3,4]
[1]University of Oxford
[2]University College London
[3]University of Edinburgh
[4]The Alan Turing Institute
`yshi@robots.ox.ac.uk`

## Abstract

Multimodal learning for generative models often refers to the learning of abstract concepts from the *commonality* of information in multiple modalities, such as vision and language. While it has proven effective for learning generalisable representations, the training of such models often requires a large amount of "related" multimodal data that shares commonality, which can be expensive to come by. To mitigate this, we develop a novel contrastive framework for generative model learning, allowing us to train the model not just by the commonality between modalities, but by the *distinction* between "related" and "unrelated" multimodal data. We show in experiments that our method enables data-efficient multimodal learning on challenging datasets for various multimodal variational autoencoder (VAE) models. We also show that under our proposed framework, the generative model can accurately identify related samples from unrelated ones, making it possible to make use of the plentiful unlabeled, unpaired multimodal data.

## 1 Introduction

To comprehensively describe concepts in the real world, humans collect multiple perceptual signals of the same object such as image, sound, text and video. We refer to each of these media as a *modality*, and a collection of different media featuring the same underlying concept is characterised as *multimodal* data. Learning from multiple modalities has been shown to yield more generalisable representations (Zhang et al., 2020; Guo et al., 2019; Yildirim, 2014), as different modalities are often complimentary in content while overlapping for common abstract concept.

Category: Laysan Albatross

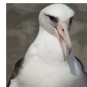
"A larger sized bird with a glowing white body and a large orange beak."

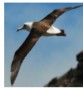
"A cigar-shaped white bird with long brown wings and hooked long bill."

Figure 1: Multimodal data from the CUB dataset

Despite the motivation, it is worth noting that the multimodal framework is not exactly data-efficient—constructing a suitable dataset requires a lot of "annotated" unimodal data, as we need to ensure that each multimodal pair is related in a meaningful way. The situation is worse when we consider more complicated multimodal settings such as language–vision, where one-to-one or one-to-many correspondence between instances of the two datasets are required, due to the difficulty in categorising data such that commonality amongst samples is preserved within categories. See Figure 1 for an example from the CUB dataset (Welinder et al., a); although the same species of bird is featured in both image-caption pairs, their content differs considerably. It would be unreasonable to apply the caption from one to describe the bird depicted in the other, necessitating one-to-one correspondence between images and captions.

However, the scope of multimodal learning has been limited to leveraging the commonality between "related" pairs, while largely ignoring "unrelated" samples potentially available in any multimodal dataset—constructed through random pairing between modalities (Figure 3). We posit that if a distinction can be established between the "related" and "unrelated" observations within a multimodal dataset, we could greatly reduce the amount of related data required for effective learning. Figure 2 formalises this proposal. Multimodal generative models in previous work (Figure 2a) typically assumes one latent variable $z$ that *always* generates related multimodal pair $(x, y)$. In this work (Figure 2b), we introduce an additional Bernoulli random variable $r$ that dictates the "relatedness" between $x$ and $y$ through $z$, where $x$ and $y$ are related when $r = 1$, and unrelated when $r = 0$.

---

[*]work done while at Oxford

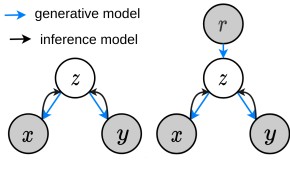

(a) Previous    (b) Ours

Figure 2: Graphical models for multimodal generative process.

While $r$ can encode different dependencies, here we make the simplifying assumption that the Pointwise Mutual Information (PMI) between $x$ and $y$ should be high when $r = 1$, and low when $r = 0$. Intuitively, this can be achieved by adopting a max-margin metric. We therefore propose to train the generative moels with a novel contrastive-style loss (Hadsell et al., 2006; Weinberger et al., 2005), and demonstrate the effectiveness of our proposed method from a few different perspectives: **Improved multimodal learning:** showing improved multimodal learning for various state-of-the-art multimodal generative models on two challenging multimodal datasets. This is evaluated on four different metrics (Shi et al., 2019) summarised in § 4.2; **Data efficiency:** learning generative models under the contrastive framework requires only 20% of the data needed in baseline methods to achieve similar performance—holding true across different models, datasets and metrics; **Label propagation:** the contrastive loss encourages a larger discrepency between related and unrelated data, making it possible to directly identify related samples using the PMI between observations. We show that these data pairs can be used to further improve the learning of the generative model.

## 2 RELATED WORK

**Contrastive loss**    Our work aims to encourage data-efficient multimodal generative-model learning using a popular *representation learning* metric—contrastive loss (Hadsell et al., 2006; Weinberger et al., 2005). There has been many successful applications of contrastive loss to a range of different tasks, such as contrastive predictive coding for time series data (van den Oord et al., 2018), image classification (Hénaff, 2020), noise contrastive estimation for vector embeddings of words (Mnih and Kavukcuoglu, 2013), as well as a range of frameworks such as DIM (Hjelm et al., 2019), MoCo (He et al., 2020), SimCLR (Chen et al., 2020) for more general visual-representation learning. The features learned by contrastive loss also perform well when applied to different downstream tasks — their ability to generalise is further analysed in (Wang and Isola, 2020) using quantifiable metrics for alignment and uniformity.

Contrastive methods have also been employed under a generative-model setting, but typically on generative adversarial networks (GANs) to either preserve or identify factors-of-variations in their inputs. For instance, SiGAN (Hsu et al., 2019) uses a contrastive loss to preserve identity for face-image hallucination from low-resolution photos, while (Yildirim et al., 2018) uses a contrastive loss to disentangle the factors of variations in the latent code of GANs. We here employ a contrastive loss in a distinct setting of multimodal generative model learning, that, as we will show with our experiments and analyses promotes better, more robust representation learning.

**Multimodal VAEs**    We also demonstrate that our approach is applicable across different approaches to learning multimodal generative models. To do so, we first summarise past work on multimodal VAE into two categories based on the modelling choice of approximated posterior $q_\Phi(z|x,y)$:

a) **Explicit joint models**: $q_\Phi$ as single joint encoder $q_\Phi(z|x,y)$.
   Example work in this area include JMVAE (Suzuki et al.), triple ELBO (Vedantam et al., 2018) and MFM (Tsai et al., 2019). Since the joint encoder require multimodal pair $(x, y)$ as input, these approaches typically require additional modelling components and/or inference steps to deal with missing modality at test time; in fact, all three approaches propose to train unimodal VAEs on top of the joint model that handles data from each modality independently.

b) **Factorised joint models**: $q_\Phi$ as factored encoders $q_\Phi(z|x,y) = f\left(q_{\phi_x}(z|x), q_{\phi_y}(z|y)\right)$.
   This was first seen in Wu and Goodman (2018), as the MVAE model with $f$ defined as a product of experts (PoE), i.e. $q_\Phi(z|x,y) = q_{\phi_x}(z|x)q_{\phi_y}(z|y)p(z)$, allowing for cross-modality generation without extra modelling components. Particularly, the MVAE caters to settings where data was not guaranteed to be always related, and where additional modalities were, in terms of information content, subsets of a primary data source—such as images and their class labels.
   Alternately, Shi et al. (2019) explored an approach that explicitly leveraged the availability of related/paired data, motivated by arguments from embodied cognition of the world. They propose the MMVAE model, which additionally differs from the MVAE model in its choice of posterior approximation—where $f$ is modelled as the mixture of experts (MoE) of unimodal posteriors—to ameliorate shortcomings to do with precision miscalibration of the PoE. Furthermore, Shi et al.

(2019) also posit four criteria that a multimodal VAE should satisfy, which we adopt in this work to evaluate the performance of our models.

**Weakly-supervised learning**  Using generative models for label propagation (see § 4.5) is a form of weak supervision. Commonly seen approaches for weakly-supervised training with incomplete data include (1) graph-based methods (such as minimum cut) (Zhou et al., 2003; Zhu et al., 2003; Blum and Chawla, 2001), (2) low-density separation methods (Joachims, 1999; Burkhart and Shan, 2020) and (3) disagreement-based models (Blum and Mitchell, 1998; Zhou and Li, 2005; 2010). However, (1) and (2) suffers from scalability issues due to computational inefficiency and optimisation complexity, while (3) works well for many different tasks but requires training an ensemble of learners.

The use of generative models for weakly supervised learning has also been explored in Nigam et al. (2000); Miller and Uyar (1996), where labels are estimated using expectation-maximisation (EM) (Dempster et al., 1977) for instances that are unlabelled. Notably, models trained with our contrastive objective does not need EM to leverage unlabelled data (see § 4.5) to determine the relatedness of two examples we only need to compute a threshold (estimation of PMI) using the trained model.

## 3  METHODOLOGY

Given data over observations from two modalties $(\boldsymbol{x}, \boldsymbol{y})$, one can learn a multimodal VAE targetting $p_\Theta(\boldsymbol{x}, \boldsymbol{y}, \boldsymbol{z}) = p(\boldsymbol{z}) p_{\theta_x}(\boldsymbol{x}|\boldsymbol{z}) p_{\theta_y}(\boldsymbol{y}|\boldsymbol{z})$, where $p_\theta(\cdot|\boldsymbol{z})$ are deep neural networks (decoders) parametrised by $\Theta = \{\theta_x, \theta_y\}$. To maximise the joint marginal likelihood $\log p_\Theta(\boldsymbol{x}, \boldsymbol{y})$, one approximates the intractable model posterior $p_\Theta(\boldsymbol{z}|\boldsymbol{x}, \boldsymbol{y})$ with a variational posterior $q_\Phi(\boldsymbol{z}|\boldsymbol{x}, \boldsymbol{y})$, allowing us to optimise a variational evidence lower bound (ELBO), defined as

$$\log p_\Theta(\boldsymbol{x}, \boldsymbol{y}) \geq \mathbb{E}_{\boldsymbol{z} \sim q_\Phi(\boldsymbol{z}|\boldsymbol{x}, \boldsymbol{y})} \left[ \log \frac{p_\Theta(\boldsymbol{z}, \boldsymbol{x}, \boldsymbol{y})}{q_\Phi(\boldsymbol{z} \mid \boldsymbol{x}, \boldsymbol{y})} \right] = \text{ELBO}(\boldsymbol{x}, \boldsymbol{y}). \tag{1}$$

This leaves open the question of how to model the approximatd posterior $q_\Phi(\boldsymbol{z}|\boldsymbol{x}, \boldsymbol{y})$. As mentioned in § 2, there are two schools of thinking, namely *explicit joint model* such as JMVAE (Suzuki et al.) and *factorised joint model* including MVAE (Wu and Goodman, 2018) and MMVAE (Shi et al., 2019). In this work we demonstrate the effectiveness of our approach across all these models.

### 3.1  CONTRASTIVE LOSS FOR "RELATEDNESS" LEARNING

Where prior work always assumes $(\boldsymbol{x}, \boldsymbol{y})$ to be related, we introduce a *relatedness* variable $r$ explicitly capturing this aspect. Our approach is motivated by the characteristics of pointwise mutual information (PMI) between related and unrelated observations across modalities:

**Hypothesis 3.1.** *Let $(\boldsymbol{x}, \boldsymbol{y}) \sim p_\Theta(\boldsymbol{x}, \boldsymbol{y})$ be a related data pair from two modalities, and let $\boldsymbol{y}'$ denote a data point unrelated to $\boldsymbol{x}$. Then, the pointwise mutual information $I(\boldsymbol{x}, \boldsymbol{y}) > I(\boldsymbol{x}, \boldsymbol{y}')$.*

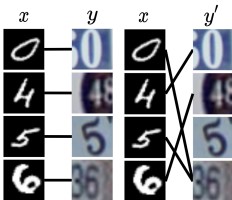

(a) Related    (b) Unrelated

Figure 3: Constructing related & unrelated samples

Note that the PMI measures the statistical dependence between values $(x, y)$, which for a joint distribution $p(x, y)$ is defined as $I(x, y) = \log \frac{p(x,y)}{p(x)p(y)}$. Hypothesis 3.1 should be a fairly uncontroversial assumption for true generative models: we say simply that under the joint distribution for related data $p_\Theta(\boldsymbol{x}, \boldsymbol{y})$, the PMI between related points $(\boldsymbol{x}, \boldsymbol{y})$ is stronger than that between unrelated points $(\boldsymbol{x}, \boldsymbol{y}')$. In fact, we demonstrate in § 4.5 that for a well-trained generative model, PMI is a good indicator of relatedness, enabling pairing of random multimodal data by thresholding the PMI estimated by the trained model. It is therefore possible to leverage relatedness while training parametrised generative model by maximising the difference in PMI between related and unrelated pairs, i.e. $I(\boldsymbol{x}, \boldsymbol{y}) - I(\boldsymbol{x}, \boldsymbol{y}')$. We show in Appendix A that this is equivalent to maximising the difference between the joint marginal likelihoods of related and unrelated pairs (Figure 3a and 3b). This casts multimodal learning as max-margin optimisation, with the contrastive (triplet) loss as a natural choice of objective: $\mathcal{L}_C(\boldsymbol{x}, \boldsymbol{y}, \boldsymbol{y}') = d(\boldsymbol{x}, \boldsymbol{y}) - d(\boldsymbol{x}, \boldsymbol{y}') + m$. Intuitively, $\mathcal{L}_C$ attempts to make distance $d$ between a positive pair $(\boldsymbol{x}, \boldsymbol{y})$ smaller than the distance between a negative pair $(\boldsymbol{x}, \boldsymbol{y}')$ by margin $m$. We adopt this loss to our objective by omitting margin $m$ and replacing $d$ with the (negative) joint marginal likelihood $-\log p_\Theta$. Following Song et al. (2016), with $N$ negative samples $\{y_i'\}_{i=1}^N$, we have

$$\mathcal{L}_C(\boldsymbol{x}, Y) = -\log p_\Theta(\boldsymbol{x}, \boldsymbol{y}) + \log \sum_{i=1}^N p_\Theta(\boldsymbol{x}, \boldsymbol{y}_i'). \tag{2}$$

We choose to put the sum over $N$ within the $\log$ following conventions in previous work on contrastive loss (van den Oord et al., 2018; Hjelm et al., 2019; Chen et al., 2020). As the loss is asymmetric, one can average over $\mathcal{L}_C(\boldsymbol{y}, X)$ and $\mathcal{L}_C(\boldsymbol{x}, Y)$ to ensure negative samples in both modalities are accounted for—giving our contrastive objective:

$$\mathcal{L}_C(\boldsymbol{x}, \boldsymbol{y}) = \frac{1}{2}\{\mathcal{L}_C(\boldsymbol{x}, Y) + \mathcal{L}_C(\boldsymbol{y}, X)\} = \underbrace{-\log p_\Theta(\boldsymbol{x}, \boldsymbol{y})}_{①} + \underbrace{\frac{1}{2}\left(\log\sum_{\boldsymbol{x}'_i=1}^{N} p_\Theta(\boldsymbol{x}', \boldsymbol{y}) + \log\sum_{\boldsymbol{y}'_i=1}^{N} p_\Theta(\boldsymbol{x}, \boldsymbol{y}')\right)}_{②} \quad (3)$$

Note that since only the joint marginal likelihood terms are needed in (3), the contrastive learning framework can be directly applied to *any* multimodal generative model without needing extra components. We also show our contrastive framework applies when the number of modalities is more than two (cf. Appendix C).

**Dissecting $\mathcal{L}_C$**   Although similar to the VAE (1), our objective (3) directly maximises $p_\Theta(\boldsymbol{x}, \boldsymbol{y})$ in ①. $\mathcal{L}_C$ by itself is not a effective objective as ② in $\mathcal{L}_C$ *minimises* $p_\Theta(\boldsymbol{x}, \boldsymbol{y})$, which can overpower the effect of ① during training.

Figure 4: $\log p(\boldsymbol{x}, \boldsymbol{y})$ of imitation, unrelated digits and random noise, where $p = \mathcal{N}(\boldsymbol{m_x}, \boldsymbol{m_y})$.

We intuit this phenomenon using a simple example in Figure 4 with natural images, showing the log likelihood of $\log p(\boldsymbol{x}, \boldsymbol{y})$ in column 2, 3, 4 (green) on a Gaussian distribution $\mathcal{N}((\boldsymbol{m_x}, \boldsymbol{m_y}); c)$, with the images in the first column of Figure 4 as means and with constant variance $c$. While achieving high $\log p(\boldsymbol{x}, \boldsymbol{y})$ requires matching $(\boldsymbol{m_x}, \boldsymbol{m_y})$ (col 2), we see both unrelated digits (col 3) and noise (col 4) can lead to (comparatively) poor joint log likelihoods. This indicates that the generative model need not generate valid, unrelated images to minimise ②—generating noise would have roughly the same effect on log likelihood. As a result, the model can trivially minimise $\mathcal{L}_C$ by generating noise that minimises ② instead of accurate reconstruction that maximises ①.

This learning dynamic is verified empirically, as we show in Figure 9 in Appendix D: optimising $\mathcal{L}_C$ by itself results in the loss approaching 0 within the first 100 iterations, and both ① and ② takes on extremely low values, resulting in a model that generates random noise.

**Final objective**   To mitigate this issue, we need to ensure that minimising ② does not overpower maximising ①. We hence introduce a hyperparameter $\gamma$ on ① to upweight the maximisation of the marginal likelihood, with the final objective to minimise

$$\mathcal{L}_C(\boldsymbol{x}, \boldsymbol{y}) = \underbrace{-\gamma \log p_\Theta(\boldsymbol{x}, \boldsymbol{y})}_{①} + \underbrace{\frac{1}{2}\left(\log\sum_{\boldsymbol{x}'_i=1}^{N} p_\Theta(\boldsymbol{x}', \boldsymbol{y}) + \log\sum_{\boldsymbol{y}'_i=1}^{N} p_\Theta(\boldsymbol{x}, \boldsymbol{y}')\right)}_{②}, \qquad \gamma > 1. \quad (4)$$

We conduct ablation studies on the effect of $\gamma$ in Appendix E, noting that larger $\gamma$ encourages better quality of generation and more stable training in some cases, while models trained with smaller $\gamma$ are better at predicting "relatedness" between multimodal samples. We also note that optimising (4) maximises the pointwise mutual information $I(\boldsymbol{x}, \boldsymbol{y})$; see Appendix B for a proof.

### 3.2  Optimising the objective

Since in VAEs we do not have access to the exact marginal likelihood, we have to optimise an approximated version of the true contrastive objective in (4). In (5) we list a few possible candidates of estimators and their relationships to the (log) joint marginal likelihood:

$$\text{ELBO} \leq \underbrace{\mathbb{E}_{\{\boldsymbol{z}_k\}_1^K \sim q_\Phi}\left[\log\frac{1}{K}\sum_{k=1}^{K}\frac{p_\Theta(\boldsymbol{z}_k, \boldsymbol{x}, \boldsymbol{y})}{q_\Phi(\boldsymbol{z}_k \mid \boldsymbol{x}, \boldsymbol{y})}\right]}_{\text{IWAE}} \leq \log p_\Theta(\boldsymbol{x}, \boldsymbol{y}) \leq \underbrace{\mathbb{E}_{\{\boldsymbol{z}_k\}_1^K \sim q_\Phi}\left[\log\sqrt[2]{\frac{1}{K}\sum_{k=1}^{K}\left(\frac{p_\Theta(\boldsymbol{z}_k, \boldsymbol{x}, \boldsymbol{y})}{q_\Phi(\boldsymbol{z}_k \mid \boldsymbol{x}, \boldsymbol{y})}\right)^2}\right]}_{\text{CUBO}} \quad (5)$$

with the ELBO from Eq. (1), the importance weighted autoencoder (IWAE), a $K$-sample lower bound estimator that can compute an arbitrarily tight bound with increasing $K$ (Burda et al., 2016), and the $\chi$ upper bound (CUBO), an *upper-bound* estimator (Dieng et al., 2017).

We now discuss our choice of approximation of $\log p_\Theta(\boldsymbol{x}, \boldsymbol{y})$ for each term in equation (4):

① Minimising this term maximises the joint marginal likelihood, which can hence be approximated with a valid lower-bound estimator (5); the IWAE estimator being the preferred choice.

② For this term, we consider two choices of estimators:

1) **CUBO**: Since ② of (4) *minimises* the joint marginal likelihoods, to ensure an upper-bound estimator to $\mathcal{L}_C$ in (4), we need to employ an upper-bound estimator. Here we propose to use CUBO in (5). While such a bounded approximation is indeed desirable, existing upper-bound estimators tend to have rather large bias/variance and can therefore yield poor quality approximations.

2) **IWAE**: We therefore also propose to estimate ② with the IWAE estimator, as it provides an *arbitrarily tight, low variance lower-bound* (Burda et al., 2016) to $\log p_\Theta(\boldsymbol{x}, \boldsymbol{y})$. Although this no longer ensure a valid bound on the objective, we hope that having a more accurate approximation to the marginal likelihood (and by extension, the contrastive loss) can affect the performance of the model positively.

We report results using both IWAE and CUBO estimators in § 4, denoted `cI` and `cC` respectively.

## 4 EXPERIMENTS

As stated in § 1, we analyse the suitability of contrastive learning for multimodal generative models from three persepctives—*improved multimodal learning* (§ 4.3), *data efficiency* (§ 4.4) and *label propagation* (§ 4.5). Throughout the experiments, we take $N = 5$ negative samples for the contrastive objective, set $\gamma = 2$ based on analyses of ablations in appendix E, and take $K = 30$ samples for our IWAE estimators. We now introduce the datasets and metrics used for our experiments.

### 4.1 DATASETS

**MNIST-SVHN**    The dataset is designed to separate conceptual complexity, i.e. digit, from perceptual complexity, i.e. color, style, size. Each data pair contains 2 samples of the same digit, one from each dataset (see examples in Figure 3a). We construct the dataset such that each instance from one dataset is paired with 30 instances of the same digit from the other dataset. Although both datasets are simple and well-studied, the many-to-many pairing between samples creates matching of different writing styles vs. backgrounds and colors, making it a challenging multimodal dataset.

**CUB Image-Captions**    We also consider a more challenging language-vision multimodal dataset, Caltech-UCSD Birds (CUB) (Welinder et al., b; Reed et al., 2016). The dataset contains 11,788 photos of birds, paired with 10 captions describing the bird's physical characteristics, collected through Amazon Mechanical Turk (AMT). See CUB image-caption pair in Figure 1.

### 4.2 METRICS

Shi et al. (2019) proposed four criteria for multimodal generative models (Figure 5, left), that we summarise and unify as metrics to evaluate these criteria for different generative models (Figure 5, right). We now introduce each criterion and its corresponding metric in detail.

**(a) Latent accuracy (Figure 5a)**    *Criterion: latent space factors into "shared" and "private" subspaces across modalities.* We fit a linear classifier on the samples from $\boldsymbol{z} \sim q_\Phi(\boldsymbol{z}|\boldsymbol{x}, \boldsymbol{y})$ to classify the information shared between the two modalities. For MNIST-SVHN, this can be the digit label as shown in Figure 5a (right). We check if $\hat{l}_z$ is the same as the digit label of the original inputs $\boldsymbol{x}$ and $\boldsymbol{y}$, with the intuition that extracting the commonality between $\boldsymbol{x}$ and $\boldsymbol{y}$ from latent representation using a linear transform, supports the claim that the latent space has factored as desired.

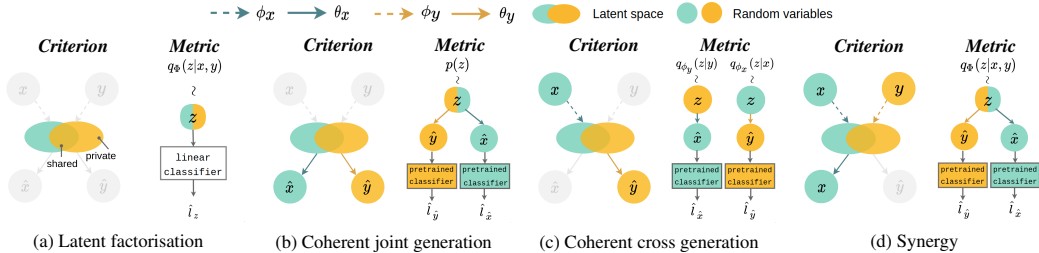

Figure 5: *Left of each pair:* Four criteria for multi-modal generative models; image adapted from Shi et al. (2019). *Right of each pair:* Four metrics to evaluate the model's performance on criterion in corresponding row.

**(b) Joint coherence (Figure 5b)** *Criterion: model generates paired samples that preserves the commonality observed in data.* Again taking MNIST-SVHN as an example, this can be verified by taking pre-trained MNIST and SVHN digit classifiers and applying them on the multimodal observations generated from the same prior sample $z$. Coherence is computed by how often generations $\hat{x}$ and $\hat{y}$ classify to the same digit, i.e. whether $\hat{l}_{\hat{x}} = \hat{l}_{\hat{y}}$.

**(c) Cross coherence (Figure 5c)** *Criterion: model generates data in one modality conditioned on the other while preserving shared commonality.* To compute cross coherence, we generate observations $\hat{x}$ using latent from unimodal marginal posterior $z \sim q_\phi(z|y)$ and $\hat{y}$ from $z \sim q_\phi(z|x)$. Similar to joint coherence, the criterion here is evaluated by predicting the label of the cross-generated samples $\hat{x}$, $\hat{y}$ using off-the-shelf MNIST and SVHN classifiers. In Figure 5c (right), the cross coherence is the frequency of which $\hat{l}_{\hat{x}} = l_y$ and $\hat{l}_{\hat{y}} = l_x$.

**(d) Synergy coherence (Figure 5d)** *Criterion: models learnt across multiple modalities should be no worse than those learnt from just one.* For consistency, we evaluate this criterion from a coherence perspective. Given generations $\hat{x}$ and $\hat{y}$ from $z \sim q_\Phi(z|x, y)$, we again examine if generated and original labels match; i.e. if $\hat{l}_{\hat{x}} = l_y = \hat{l}_{\hat{y}} = l_x$.

See Appendix F for details on architecture. All quantitative results are reported over 5 runs. In addition to these quantitative metrics, we also showcase the qualitative results on both datasets in Appendix G and Appendix H.

## 4.3 IMPROVED MULTIMODAL LEARNING

***Finding:*** *Contrastive learning improves multimodal learning across all models and datasets.*

**MNIST-SVHN** See Table 1 (*top, 100% data used*) for results on the full MNIST-SVHN dataset. Note that for MMVAE, since the joint posterior $q_\Phi$ is factorised as the mixture of unimodal posteriors $q_{\phi_x}$ and $q_{\phi_y}$, the model never directly takes sample from the explicit form of the joint posterior. Instead, it takes equal number of samples from each unimodal posteriors, reflective of the equal weighting of the mixture. As a result, it is not meaningful to compute synergy coherence for MMVAE as it is exactly the same as the coherence of any single-way generation.

From Table 1 (*top, 100% data used*), we see that models trained on our contrastive objective (cI-<MODEL> and cC-<MODEL>) improves multimodal learning performance significantly for all three generative models evaluated on the metrics. The results showcase the robustness of our approach from the perspectives of modelling choice and metric of interests. In particular, note that for the best performing model MMVAE, using IWAE estimator for ② of Eq (4) (cI-MMVAE) yields slightly better results than CUBO (cC-MMVAE), while for the two other models the performance for different estimators are similar.

We also include qualitative results for MNIST-SVHN, including generative results, marginal likelihood table and diversity analysis in Appendix G.

Table 1: Evaluation of baselines MMVAE, MVAE, JMVAE and their contrastive variations (cI-<MODEL>, cC-<MODEL> for the IWAE and CUBO estimators used in Eq (4), respectively), on MNIST(M)-SVHN(S) dataset, using 100% (top) and 20% (bottom) of data.

| Data | Methods | Latent accuracy (%) | | Joint Coherence (%) | Cross coherence (%) | | Synergy coherence (%) | |
|---|---|---|---|---|---|---|---|---|
| | | M | S | | S→M | M→S | joint→M | joint→S |
| | MMVAE | 92.48 ($\pm$0.37) | 79.03 ($\pm$1.17) | 42.32 ($\pm$2.97) | 70.77 ($\pm$0.35) | 85.50 ($\pm$1.05) | — | — |
| | **cI-MMVAE** | **93.97** ($\pm$0.36) | **81.87** ($\pm$0.52) | 43.94 ($\pm$0.96) | **79.66** ($\pm$0.59) | **92.67** ($\pm$0.29) | — | — |
| | cC-MMVAE | 93.10 ($\pm$0.17) | 80.88 ($\pm$0.80) | **45.46** ($\pm$0.78) | 79.34 ($\pm$0.54) | 92.35 ($\pm$0.46) | — | — |
| 100% of data used | MVAE | 91.65 ($\pm$0.17) | 64.12 ($\pm$4.58) | 9.42 ($\pm$7.82) | 10.98 ($\pm$0.56) | 21.88 ($\pm$2.21) | 64.60 ($\pm$9.25) | 52.91 ($\pm$8.11) |
| | cI-MVAE | 96.97 ($\pm$0.84) | 75.94 ($\pm$6.20) | 15.23 ($\pm$10.46) | 10.85 ($\pm$1.17) | 27.70 ($\pm$2.09) | 85.07 ($\pm$7.73) | 75.67 ($\pm$4.13) |
| | cC-MVAE | 97.42 ($\pm$0.40) | 81.07 ($\pm$2.03) | 8.85 ($\pm$3.86) | 12.83 ($\pm$2.25) | 30.03 ($\pm$2.46) | 75.25 ($\pm$5.31) | 69.42 ($\pm$3.94) |
| | JMVAE | 84.45 ($\pm$0.87) | 57.98 ($\pm$1.27) | 42.18 ($\pm$1.50) | 49.63 ($\pm$1.78) | 54.98 ($\pm$3.02) | 85.77 ($\pm$0.66) | 68.15 ($\pm$1.38) |
| | cI-JMVAE | 84.58 ($\pm$1.49) | 64.42 ($\pm$1.42) | 48.95 ($\pm$2.31) | 58.16 ($\pm$1.83) | 70.61 ($\pm$3.13) | **93.45** ($\pm$0.52) | 84.00 ($\pm$0.97) |
| | cC-JMVAE | 83.67 ($\pm$3.48) | 66.64 ($\pm$2.92) | 47.27 ($\pm$4.52) | 59.73 ($\pm$3.85) | 69.49 ($\pm$2.19) | 91.21 ($\pm$6.59) | **84.07** ($\pm$3.19) |
| | MMVAE | 88.54 ($\pm$0.37) | 68.90 ($\pm$1.79) | 37.71 ($\pm$0.60) | 59.52 ($\pm$0.28) | 76.33 ($\pm$2.23) | — | — |
| | **cI-MMVAE** | 91.64 ($\pm$0.06) | **73.02** ($\pm$0.80) | **42.74** ($\pm$0.36) | **69.51** ($\pm$1.18) | **86.75** ($\pm$0.28) | — | — |
| | cC-MMVAE | 92.10 ($\pm$0.19) | 71.29 ($\pm$1.05) | 40.77 ($\pm$0.93) | 68.43 ($\pm$0.90) | 86.24 ($\pm$0.89) | — | — |
| 20% of data used | MVAE | 90.29 ($\pm$0.57) | 33.44 ($\pm$0.26) | 10.88 ($\pm$9.15) | 8.72 ($\pm$0.92) | 12.12 ($\pm$3.38) | 42.10 ($\pm$5.22) | 44.95 ($\pm$5.92) |
| | cI-MVAE | **93.72** ($\pm$1.09) | 56.74 ($\pm$7.97) | 12.79 ($\pm$6.82) | 14.18 ($\pm$2.19) | 20.23 ($\pm$4.55) | 75.36 ($\pm$5.05) | 64.81 ($\pm$4.81) |
| | cC-MVAE | 92.74 ($\pm$2.97) | 52.99 ($\pm$8.35) | 17.95 ($\pm$12.52) | 14.70 ($\pm$1.65) | 24.90 ($\pm$5.77) | 56.86 ($\pm$18.84) | 54.28 ($\pm$9.86) |
| | JMVAE | 77.53 ($\pm$0.13) | 52.55 ($\pm$2.18) | 26.37 ($\pm$0.54) | 42.58 ($\pm$5.32) | 41.44 ($\pm$2.26) | 85.07 ($\pm$9.74) | 51.95 ($\pm$2.28) |
| | cI-JMVAE | 77.57 ($\pm$4.02) | 57.91 ($\pm$1.28) | 32.58 ($\pm$5.89) | 51.85 ($\pm$1.27) | 47.92 ($\pm$10.32) | **92.54** ($\pm$1.13) | 67.01 ($\pm$8.72) |
| | cC-JMVAE | 81.11 ($\pm$2.76) | 57.85 ($\pm$2.23) | 34.00 ($\pm$7.18) | 50.73 ($\pm$0.45) | 56.89 ($\pm$6.18) | 88.36 ($\pm$4.38) | **68.49** ($\pm$8.82) |

**CUB** Following Shi et al. (2019), for the images in CUB, we observe and generate in feature space instead of pixel space by preprocessing the images using a pre-trained ResNet-101 (He et al., 2016). A nearest-neighbour lookup among all the features in the test set is used to project the feature generations of the model back to image space. This helps circumvent CUB image complexities to some extent—as the primary goal here is to learn good models and representations of multimodal data, rather than a focus on pixel-level image quality of generations.

The metrics listed in § 4.2 can also be applied to CUB with some modifications. Since bird-species classes are disjoint for the train and test sets, and as we show in Figure 1 contains substantial in-class variance, it is not constructive to evaluate these metrics using bird categories as labels. In Shi et al. (2019), the authors propose to use Canonical Correlation Analysis (CCA)—used by Massiceti et al. as a reliable vision-language correlation baseline—to compute coherence scores between generated image-caption pairs; which we employ (i.e. (b), (c), (d) in Figure 5) for CUB.

We show the results in Table 2 (*top, 100% data used*). We see that our contrastive approach (both cI-`<MODEL>` and cC-`<MODEL>`) is even more effective on this challenging vision-language dataset, with significant improvements to the correlation of generated image-caption pairs. It is also on this more complicated dataset where the advantages of using the stable, low-variance IWAE estimator are highlighted — for both MVAE and JMVAE, the contrastive objective with CUBO estimator suffers from numerical stability issues, yielding close-to-zero correlations for all metrics in Table 2. Results for these models are therefore omitted.

We also show the qualitative results for CUB in Appendix H.

Table 2: Evaluation of baseline MMVAE, MVAE, JMVAE and their contrastive variations (cI-`<MODEL>` and cC-`<MODEL>`) on CUB image(`img`)-caption(`cap`) dataset, using 100% (top) and 20% (bottom) of data.

| Data | Methods | Joint Coherence (CCA) | Cross coherence (CCA) | | Synergy coherence (CCA) | |
|---|---|---|---|---|---|---|
| | | | img→cap | cap→img | joint→img | joint→cap |
| 100% of data used | MMVAE | 0.212 ($\pm$2.94e-2) | 0.154 ($\pm$7.05e-3) | 0.244 ($\pm$5.83e-3) | - | - |
| | cI-MMVAE | **0.314** ($\pm$3.12e-2) | 0.188 ($\pm$4.02e-3) | 0.334 ($\pm$1.20e-2) | - | - |
| | **cC-MMVAE** | 0.263 ($\pm$1.47e-2) | 0.244 ($\pm$2.24e-2) | **0.369** ($\pm$3.18e-3) | - | - |
| | MVAE | 0.075 ($\pm$8.71e-2) | -0.008 ($\pm$1.41e-4) | 0.000 ($\pm$1.84e-3) | -0.002 ($\pm$4.95e-3) | 0.001 ($\pm$1.13e-3) |
| | cI-MVAE | 0.209 ($\pm$3.12e-2) | 0.247 ($\pm$1.12e-4) | -0.008 ($\pm$3.99e-4) | **0.218** ($\pm$5.27e-3) | 0.148 ($\pm$9.23e-3) |
| | JMVAE | 0.220 ($\pm$1.19e-2) | 0.157 ($\pm$4.98e-2) | 0.191 ($\pm$3.11e-2) | 0.212 ($\pm$1.23e-2) | 0.143 ($\pm$1.14e-1) |
| | cI-JMVAE | 0.255 ($\pm$3.24e-3) | 0.149 ($\pm$1.25e-3) | 0.226 ($\pm$7.48e-2) | 0.202 ($\pm$1.12e-4) | **0.176** ($\pm$6.23e-2) |
| 20% of data used | MMVAE | 0.117 ($\pm$1.51e-2) | 0.094 ($\pm$7.21e-3) | 0.153 ($\pm$1.47e-2) | - | - |
| | cI-MMVAE | 0.206 ($\pm$1.65e-2) | 0.136 ($\pm$1.51e-2) | 0.251 ($\pm$2.39e-2) | - | - |
| | cC-MMVAE | 0.226 ($\pm$4.69e-2) | 0.188 ($\pm$2.80e-2) | **0.273** ($\pm$8.67e-3) | - | - |
| | MVAE | 0.091 ($\pm$2.63e-2) | 0.008 ($\pm$4.81e-3) | 0.005 ($\pm$7.50e-3) | 0.020 ($\pm$8.77e-3) | 0.009 ($\pm$1.17e-2) |
| | cI-MVAE | 0.132 ($\pm$3.33e-2) | **0.192** ($\pm$3.91e-2) | -0.002 ($\pm$3.61e-3) | 0.162 ($\pm$7.89e-2) | 0.081 ($\pm$3.82e-2) |
| | JMVAE | 0.127 ($\pm$3.76e-2) | 0.118 ($\pm$3.82e-3) | 0.154 ($\pm$8.34e-3) | 0.181 ($\pm$1.26e-2) | 0.139 ($\pm$1.33e-2) |
| | **cI-JMVAE** | **0.269** ($\pm$1.20e-2) | 0.134 ($\pm$4.24e-4) | 0.210 ($\pm$2.35e-2) | **0.192** ($\pm$1.41e-4) | **0.168** ($\pm$3.82e-3) |

## 4.4 DATA EFFICIENCY

*Finding: Contrastive learning on 20% of data matches baseline models on full data.*

We plot the quantitative performance of MMVAE with and without contrastive learning against the percentage of the original dataset used, as seen in Figure 6. We observe that the performance of contrastive MMVAE with the IWAE (*cI-MMVAE, red*) and CUBO estimators (*cC-MMVAE, yellow*) are *consistently* better than the baselines (*MMVAE, blue*), and that baseline performance using *all* related data is matched by the contrastive MMVAE using just 10—20% of data. The partial datasets used here are constructed by first taking $n\%$ of each unimodal dataset, then pairing to create multimodal datasets (§ 4.1)—ensuring it contains the requisite amount of "related" samples. In addition, we reproduce results generated from using 100% of the data in MNIST-SVHN and CUB (Tables 1 and 2) using only 20% of the original multimodal datasets (Tables 1 and 2 (*bottom*)). Comparing results between *top* vs. *bottom* in Tables 1 and 2 shows that this finding holds across the models, on both MNIST-SVHN and CUB datasets. This shows that the efficiency gains from a contrastive approach is invariant to VAE type, data, and metrics used, underscoring its effectiveness.

## 4.5 LABEL PROPAGATION

*Finding: Generative models learned contrastively are good predictors of "relatedness", enabling label propagation and matching baseline performance on full datasets, using only 10% of data.*

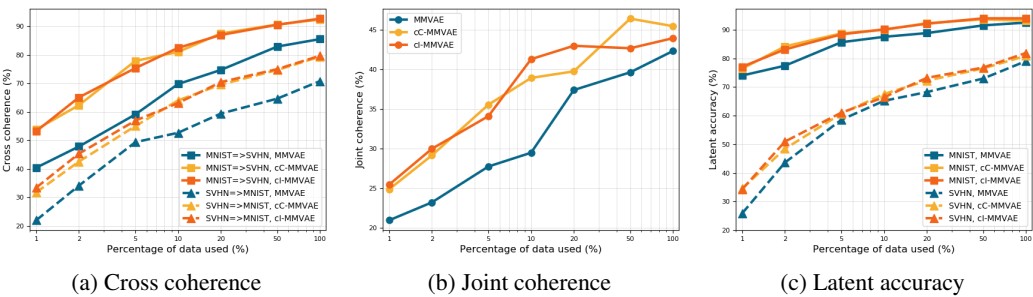

|  |  |  |
|:---:|:---:|:---:|
| (a) Cross coherence | (b) Joint coherence | (c) Latent accuracy |

Figure 6: Performance of **MMVAE**, **cI-MMVAE** and **cC-MMVAE** using $n\%$ of MNIST-SVHN.

Here, we show that our contrastive framework encourages a larger discrepancy between the PMI of related vs. unrelated data, as set out in hypothesis 3.1, allowing one to first train the model on a small subset of related data, and subsequently construct a classifier using PMI that identifies related samples in the remaining data. We now introduce our pipeline for label propagation in details.

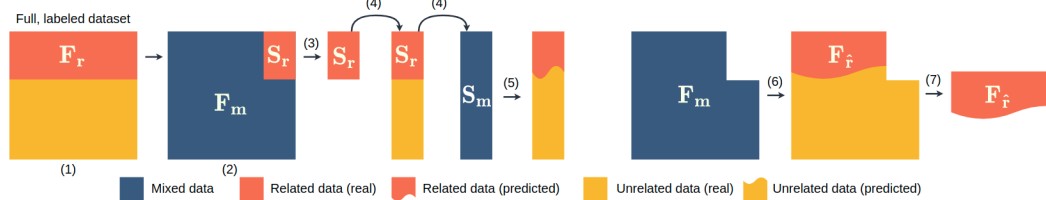

Figure 7: Pipeline of label propagation

**Pipeline**  As showing in Figure 7, we first construct a full dataset by randomly matching instances in MNIST and SVHN, and denote the related pairs by $F_r$ (full, related). We further assume access to only $n\%$ of $F_r$, denoted as $S_r$ (small, related), and denote the rest as $F_m$, containing a mix of related and unrelated pairs. Next, we train a generative model $g$ on $S_r$. To find a relatedness threshold, we construct a small, mixed dataset $S_m$ by randomly matching samples across modalities in $S_r$. Given relatedness ground-truth for $S_m$, we can compute the PMI $I(\boldsymbol{x}, \boldsymbol{y}) = \log p_\Theta(\boldsymbol{x}, \boldsymbol{y}) - \log p_{\theta_x}(\boldsymbol{x}) p_{\theta_y}(\boldsymbol{y})$ for all pairs $(\boldsymbol{x}, \boldsymbol{y})$ in $S_m$ and estimate an optimal threshold. This threshold can now be applied to the full, mixed dataset $F_m$ to identify related pairs giving us a new related dataset $F_{\hat{r}}$, which can be used to further improve the performance of the generative model $g$.

**Results**  In Figure 8 (a-e), we plot the performance of baseline MMVAE (blue) and contrastive MMVAE with IWAE (cI-MMVAE, red) and CUBO (cC-MMVAE, yellow) estimators for term ② of Eq (4), trained with (solid curves) and without (dotted curves) label propagation. Here, the x-axis is the proportion in size of $S_r$ to $F_r$, i.e. the percentage of related data used to pretrain the generative model before label propagation. We compare these results to MMVAE trained on all related data $F_r$ (cyan horizontal line) as a "best case scenario" of these training regimes.

Clearly, label propagation using a contrastive model with IWAE estimator is helpful, and in general the improvement is greater when less data is available; Figure 8 also shows that when $S_r$ is 10% of $F_r$, cI-MMVAE is competitive with the performance of baseline MMVAE trained on $F_r$.

For baselines MMVAE and cC-MMVAE however, label propagation hurts performance no matter the size of $S_r$, as shown by the blue and yellow curves in Figure 8 (a-e). The advantages of cI-MMVAE is further demonstrated in Figure 8f, where we compute the precision, recall, and $F_1$ score of relatedness prediction on $F_m$, for models trained on 10% of all related data. We also compare to a simple label-propagation baseline, where the relatedness of $F_m$ is predicted using a siamese network (Hadsell et al., 2006) trained on the same 10% dataset. Notably, while the Siamese baseline in Figure 8f is a competitive predictor of relatedness, cI-MMVAE has the highest $F_1$ score amongst all four, making it the most reliable indicator of relatedness. Beyond that, note that with the contrastive MMVAE, relatedness can be predicted *without additional training* and only requires a simple threshold computation directly computed using the generative model. The fact that the cI-MMVAE's relatedness-prediction performance is the *only* one that matches the Siamese baseline strongly supports the view that the contrastive loss encourages generative models to utilise and better learn the relatedness between multimodal pairs; in addition, the poor performance of cC-MMVAE shows that the advantages of having a bounded estimation to the contrastive objective by using an

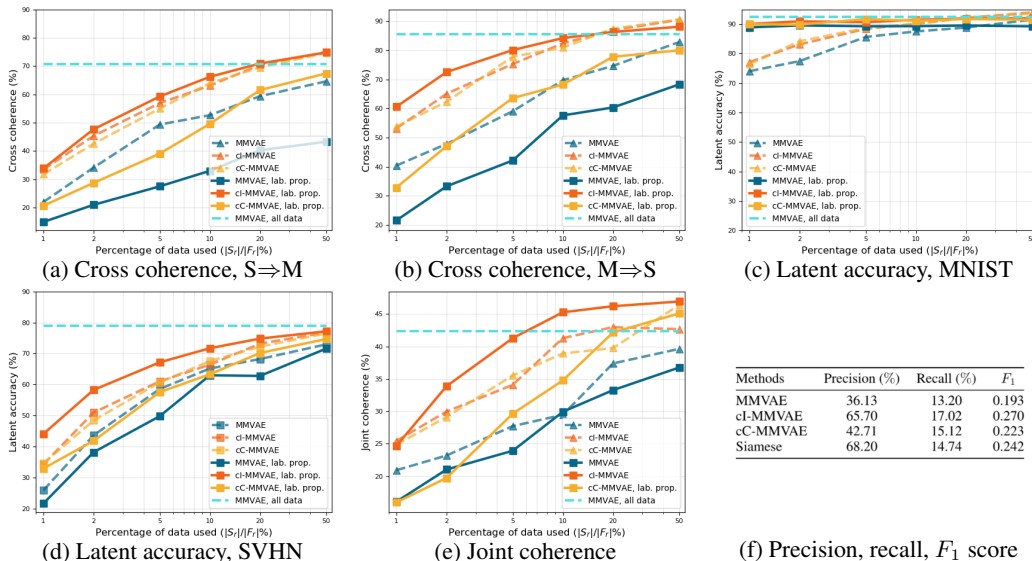

Figure 8: Models with and without label propagation using **MMVAE**, **cI-MMVAE** and **cC-MMVAE**.

upper-bound for ② is overshadowed by the high bias of CUBO, and that one may benefit more from choosing a low variance lower-bound like IWAE.

## 5 CONCLUSION

We introduced a contrastive-style objective for multimodal VAE, aiming at reducing the amount of multimodal data needed by exploiting the distinction between "related" and "unrelated" multimodal pairs. We showed that this objective improves multimodal training, drastically reduce the amount of multimodal data needed, and establishes a strong sense of "relatedness" for the generative model. These findings hold true across a multitude of datasets, models and metrics. The positive results of our method indicates that it is beneficial to utilise the relatedness information when training on multimodal data, which has been largely ignored in previous work. While we propose to utilise it implicitly through contrastive loss, one may consider relatedness as a random variable in the graphical model and see if explicit dependency on relatedness can be useful. It is also possible to extend this idea to Generative adversarial networks (GANs), by employing an additional discriminator that evaluates relatedness between generations across modalities. We will leave these interesting directions to be explored by future work.

ACKNOWLEDGEMENTS

YS and PHST were supported by the Royal Academy of Engineering under the Research Chair and Senior Research Fellowships scheme, EPSRC/MURI grant EP/N019474/1 and FiveAI. YS was additionally supported by Remarkdip through their PhD Scholarship Programme. BP is supported by the Alan Turing Institute under the EPSRC grant EP/N510129/1. Special thanks to Elise van der Pol for helpful discussions on contrastive learning.

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

# Appendix:

## A  FROM POINTWISE MUTUAL INFORMATION TO JOINT MARGINAL LIKELIHOOD

In this section, we show that for the purpose of utilising the relatedness between mutlimodal pairs, maximising the difference between *pointwise mutual information* between related points $\boldsymbol{x}, \boldsymbol{y}$ and unrelated points $\boldsymbol{x}, \boldsymbol{y}'$ is equivalent to maximising the difference between their log *joint marginal likelihoods*.

To see this, we can expand $I(\boldsymbol{x}, \boldsymbol{y}) - I(\boldsymbol{x}, \boldsymbol{y}')$ as

$$
\begin{aligned}
&I(\boldsymbol{x}, \boldsymbol{y}) - I(\boldsymbol{x}, \boldsymbol{y}') \\
&= [\log p_\Theta(\boldsymbol{x}, \boldsymbol{y}) - \log p_\Theta(\boldsymbol{x}) - \log p_\Theta(\boldsymbol{y})] - [\log p_\Theta(\boldsymbol{x}, \boldsymbol{y}') - \log p_\Theta(\boldsymbol{x}) - \log p_\Theta(\boldsymbol{y}')] \\
&= \underbrace{\log p_\Theta(\boldsymbol{x}, \boldsymbol{y}) - \log p_\Theta(\boldsymbol{x}, \boldsymbol{y}')}_{①} + \underbrace{\log p_\Theta(\boldsymbol{y}') - \log p_\Theta(\boldsymbol{y})}_{②} .
\end{aligned}
\tag{6}
$$

In (6) the PMI difference is decomposed as two terms: ① the difference between joint marginal likelihoods and ② the difference between marginal likelihoods of different instances of $\boldsymbol{y}$-s. It is clear that since ② involves only one modality and only accounts for the difference in values between $\boldsymbol{y}$ and $\boldsymbol{y}'$, it is not relevant to the relatedness of $\boldsymbol{x}, \boldsymbol{y}$.

Therefore, for the purpose of utilising relatedness information, we only need to maximise $I(\boldsymbol{x}, \boldsymbol{y}) - I(\boldsymbol{x}, \boldsymbol{y}')$ through maximising term ①, i.e. the difference between joint marginal likelihoods.

## B  CONNECTION OF FINAL OBJECTIVE TO POINTWISE MUTUAL INFORMATION

Here, we show that minimising the objective in (4) maximises the PMI between $\boldsymbol{x}$ and $\boldsymbol{y}$:

$$
\begin{aligned}
\mathcal{L}(\boldsymbol{x}, \boldsymbol{y}) &= -\gamma \log p_\Theta(\boldsymbol{x}, \boldsymbol{y}) + \frac{1}{2} \left( \log \sum_{\boldsymbol{x}' \in X} p_\Theta(\boldsymbol{x}', \boldsymbol{y}) + \log \sum_{\boldsymbol{y}' \in Y} p_\Theta(\boldsymbol{x}, \boldsymbol{y}') \right) \\
&= -(\gamma - \frac{1}{2}) \log p_\Theta(\boldsymbol{x}, \boldsymbol{y}) - \frac{1}{2} \log \frac{p_\Theta(\boldsymbol{x}, \boldsymbol{y})}{\sum_{\boldsymbol{y}' \in Y} p_\Theta(\boldsymbol{x}, \boldsymbol{y}') \sum_{\boldsymbol{x}' \in X} p_\Theta(\boldsymbol{y}, \boldsymbol{x}')} \\
&\approx -(\gamma - \frac{1}{2}) \log p_\Theta(\boldsymbol{x}, \boldsymbol{y}) - \frac{1}{2} \log \underbrace{\frac{p_\Theta(\boldsymbol{x}, \boldsymbol{y})}{p_\Theta(\boldsymbol{x}) p_\Theta(\boldsymbol{y})}}_{I(\boldsymbol{x}, \boldsymbol{y})}
\end{aligned}
\tag{7}
$$

We see in (7) that minimising $\mathcal{L}$ can be decomposed to maximising both the joint marginal likelihood $p_\Theta(\boldsymbol{x}, \boldsymbol{y})$ and an **approximation** of PMI $I(\boldsymbol{x}, \boldsymbol{y})$. Note that since $\gamma > 1$, we can be sure that the joint marginal likelihood weighting $\gamma - \frac{1}{2}$ is non-negative.

## C  GENERALISATION TO $M > 2$ MODALITIES

In this section we show how the contrastive loss generalise to cases where number of modalities considered $M$ is greater than 2.

Given observations from $M$ modalities $D = \{X_1, X_2, \cdots, X_m, \cdots, X_M\}$, where $X_m$ denotes unimodal dataset of modalit $m$ of size $N_m$, i.e. $X_m = \{\boldsymbol{x}_m^{(i)}\}_{i=1}^{N_m}$. Similar to (2), we can write the assymetrical contrastive loss for any observation $\boldsymbol{x}_m^{(i)}$ from modality $m$, where negative samples are taken for all $(M-1)$ other modalities:

$$\mathcal{L}_C(\boldsymbol{x}_m^{(i)}, D_{\tilde{m}}) = -\log p_\Theta(\boldsymbol{x}_{1:M}^{(i)}) + \log \sum_{\substack{d=1 \\ d \neq m}}^{M} \sum_{\substack{j=1 \\ j \neq i}}^{N} p_\Theta(\boldsymbol{x}_{1:(d-1),(d+1):M}^{(i)}, \boldsymbol{x}_d^{(j)}). \tag{8}$$

We can therefore rewrite (3) as:

$$\mathcal{L}_C(\boldsymbol{x}_{1:M}^{(i)}) = \frac{1}{M} \sum_{m=1}^{M} \mathcal{L}_C(\boldsymbol{x}_m^{(i)}, D_{\tilde{m}}) \tag{9}$$

$$= -\log p_\Theta(\boldsymbol{x}_{1:M}^{(i)}) + \frac{1}{M} \sum_{m=1}^{M} \left( \log \sum_{\substack{d=1 \\ d \neq m}}^{M} \sum_{\substack{j=1 \\ j \neq i}}^{N} p_\Theta(\boldsymbol{x}_{1:(d-1),(d+1):M}^{(i)}, \boldsymbol{x}_d^{(j)}) \right), \tag{10}$$

where $N$ is the number of negative samples, all $\log p_\Theta(\boldsymbol{x}_{1:M})$ are approximated by the following joint ELBO for $M$ modalities:

$$\log p_\Theta(\boldsymbol{x}_{1:M}) \geq \mathbb{E}_{\boldsymbol{z} \sim q_\Phi(\boldsymbol{z}|\boldsymbol{x}_{1:M})} \left[ \log \frac{p_\Theta(\boldsymbol{z}, \boldsymbol{x}_{1:M})}{q_\Phi(\boldsymbol{z} \mid \boldsymbol{x}_{1:M})} \right] = \mathrm{ELBO}(\boldsymbol{x}_{1:M}). \tag{11}$$

While the above gives us the true generalisation of (3), we note that the number of times where ELBO needs to be evaluated in (10) is $\mathcal{O}(M^2 N)$, making it difficult to implement this objective in practice, especially on more complicated datasets. We therefore propose a simplified version of the objective, where we estimate the second term of (10) with $N$ sets of random samples from all modalities. Specifically, we can precompute the following $M \times N$ random index matrix $J$:

$$J = \begin{bmatrix} j_{11} & j_{12} & \cdots & j_{1N} \\ j_{21} & j_{22} & \cdots & j_{2N} \\ \vdots & & & \vdots \\ j_{M1} & j_{M2} & \cdots & j_{MN} \end{bmatrix}, \tag{12}$$

where each entry of $J$ is a random integer taken from range $[1, N_m]$. We can then replace the second term of (10) random samples selected by the indices in $J$, giving us

$$\mathcal{L}_C(\boldsymbol{x}_{1:M}^{(i)}) \approx -\log p_\Theta(\boldsymbol{x}_{1:M}^{(i)}) + \left( \log \sum_{n=1}^{N} p_\Theta(\boldsymbol{x}_1^{(J_{1n})}, \boldsymbol{x}_2^{(J_{2n})}, \cdots \boldsymbol{x}_M^{(J_{Mn})}) \right). \tag{13}$$

The number of times ELBO needs to be computed is now $\mathcal{O}(N)$, and is no longer relevant to the number of modalities $M$.

We can now also generalise the final objective in (4) to $M$ modalities:

$$\mathcal{L}(\boldsymbol{x}_{1:M}^{(i)}) = -\gamma \mathrm{ELBO}(\boldsymbol{x}_{1:M}^{(i)}) + \left( \underset{n \in N}{\mathrm{logsumexp}}\, \mathrm{ELBO}(\boldsymbol{x}_1^{(J_{1n})}, \boldsymbol{x}_2^{(J_{2n})}, \cdots \boldsymbol{x}_M^{(J_{Mn})}) \right). \tag{14}$$

## D  THE INEFFECTIVENESS OF TRAINING WITH $\mathcal{L}_C$ ONLY

We demonstrate why training with the contrastive loss proposed in (3) is ineffective, and why additional ELBO term is needed for the final objective. As we show in Figure 9, when training with $\mathcal{L}_C$ only, while the contrastive loss (green) quickly drops to zero, both term ① and ② in (3) also reduces drastically. This means the joint marginal likelihood of any generation $\log p_\Theta(\boldsymbol{x}, \boldsymbol{y})$ is small regardless the relatedness of $(\boldsymbol{x}, \boldsymbol{y})$.

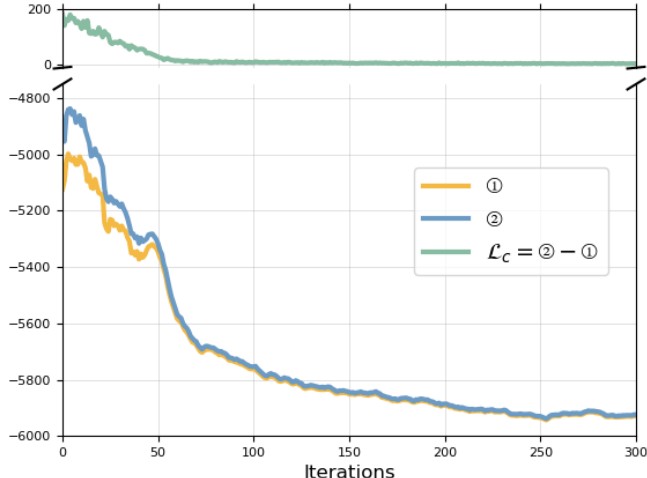

Figure 9: First 300 iterations of training using contrastive loss $\mathcal{L}_C$ only.

In comparison, we also plot the training curve for model trained on the final objective in (4), which upweights term ① in (3) by $\gamma$. We see in Figure 10 that by setting $\gamma = 2$, the joint marginal likelihood (yellow and blue curve) improves during training, while $\mathcal{L}_C$ (green curve) gets minimised.

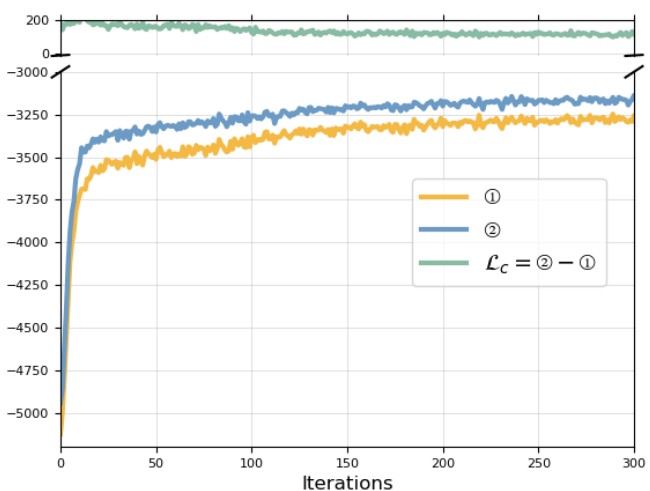

Figure 10: First 300 iterations of training with final loss $\mathcal{L}$, where $\gamma = 2$.

# E  ABLATION STUDY OF $\gamma$

In § 3, we specified that $\gamma$ needs to be greater than 1 to offset the negative effect of minimising ELBO through term ② in (4). Here, we study the effect of $\gamma$ in details.

Figure 11 compares latent accuracy, cross coherence and joint coherence of MMVAE on MNIST-SVHN dataset trained on different values of $\gamma$. Note that here we only consider cases where $\gamma \geq 1$, since the minimum value of $\gamma$ is 1. In this case, the loss reduces to the original contrastive objective in (3).

A few interesting observations from the plot are as follows: First, when $\gamma = 1$, the model is trained using the contrastive loss only, and as we showed is an ineffective objective for generative model learning. This is verified again in Figure 11 — when $\gamma = 1$, both coherences and latent accuracies take on extremely low values; interestingly, there is a significant boost of performance across all metrics by simply increasing the value of $\gamma$ from 1 to 1.1; after that, as the value of $\gamma$ increases, performance on most metrics decreases monotonically (joint coherence being the only exception), and eventually converges to baseline MMVAE (dotted lines in Figure 11). This is unsurprising, since the final objective in (4) reduces to the original joint ELBO as $\gamma$ approaches infinity.

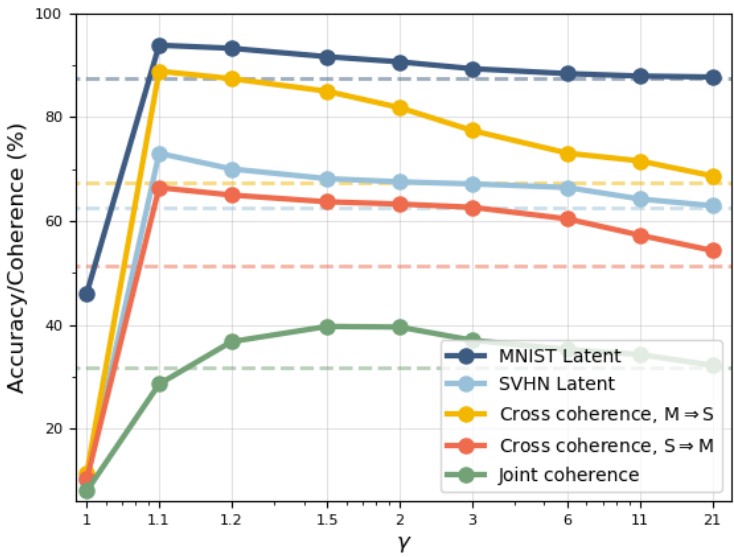

Figure 11: Performance on different metrics for different values of $\gamma$. Dotted lines represents the performance of baseline MMVAE.

Figure 11 seems to suggest that 1.1 is the optimal value for hyperparameter $\gamma$, however close inspection of the qualitative generative results shows that this might not be the case. See Figure 12 for a comparison of the model's generation between MMVAE models trained on (from left to right) $\gamma = 1.1$, $\gamma = 2$ and $\gamma = +\infty$ (i.e. original MMVAE). Although $\gamma = 1.1$ yields model with high coherence scores, we can clearly see from the left-most column of Figure 12 that the generation of the model seems deprecated, especially for the SVHN modality, where the backgrounds of model's generation appear to be unnaturally spotty and deformed. This problem is mitigated by increasing $\gamma$ — as shown in Figure 11, the image generation quality of $\gamma = 2$ (middle column) is not visibly different from that of $\gamma = +\infty$ (right column).

To verify this observation, we also compute the marginal log likelihood $\log p_\Theta(\boldsymbol{x}, \boldsymbol{y})$ to quantify the quality of generations. We compute this for all $\gamma$s considered in Figure 11, and take the average over the entire test set. From the results in Figure 13, we can see a significant increase of the log likelihood between $\gamma = 1.1$ to $\gamma = 1$. This gain in image generation quality then slows down as $\gamma$ further increases, and as all other metrics converges to the original MMVAE model.

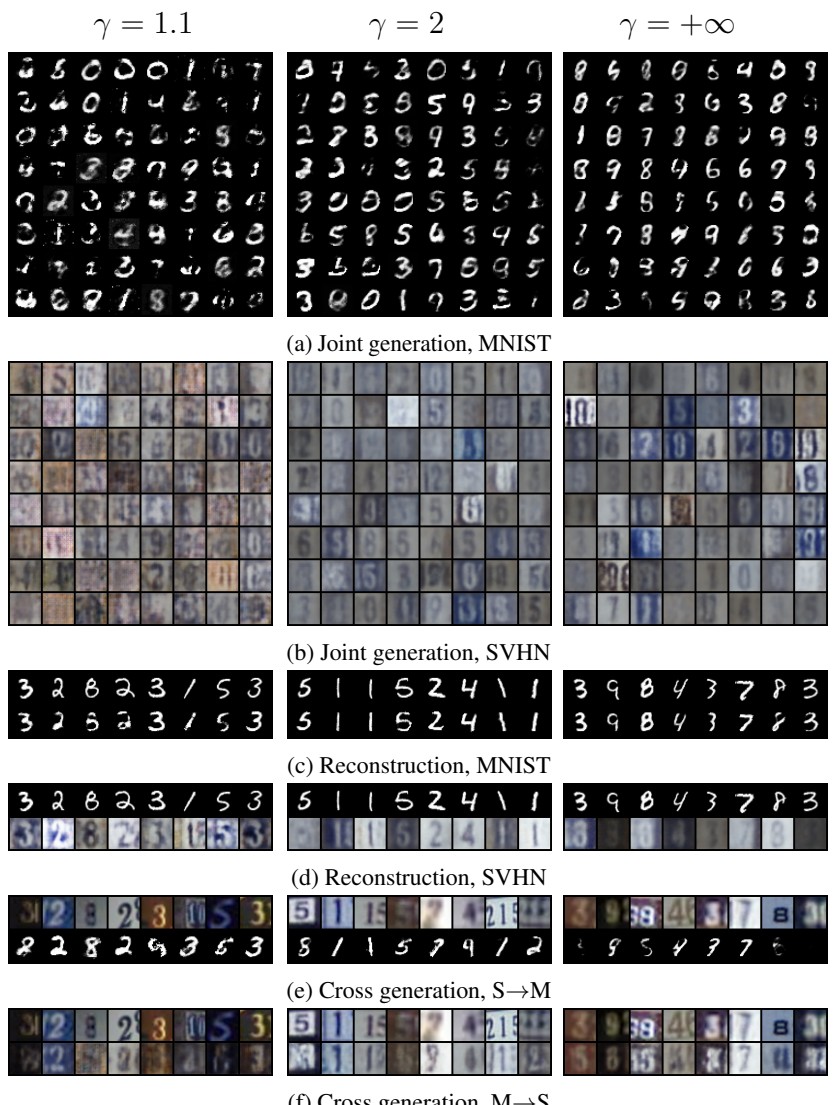

$\gamma = 1.1$  $\gamma = 2$  $\gamma = +\infty$

(a) Joint generation, MNIST

(b) Joint generation, SVHN

(c) Reconstruction, MNIST

(d) Reconstruction, SVHN

(e) Cross generation, S→M

(f) Cross generation, M→S

Figure 12: Generations of MMVAE model trained using the final contrastive objective, with (from left to right) $\gamma = 1.1$, 2 and $+\infty$. Note in (c), (d), (e), (f), the top rows are the inputs and the bottom rows are their corresponding reconstruction/cross generation.

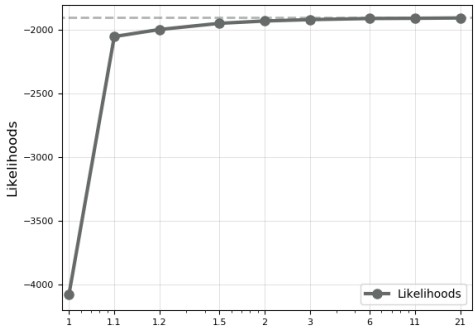

Figure 13: Performance on different metrics for different values of $\gamma$. Dotted lines represents the performance of baseline MMVAE.

# F  ARCHITECTURE

We use architectures listed in Table 3 for the unimodal encoder and decoder for MMVAE, MVAE and JMVAE. For JMVAE we use an extra joint encoder, the architecture of which is described in Table 4.

| Encoder | Decoder |
|---------|---------|
| Input $\in \mathbb{R}^{1x28x28}$ | Input $\in \mathbb{R}^L$ |
| FC. 400 ReLU | FC. 400 ReLU |
| FC. $L$, FC. $L$ | FC. 1 x 28 x 28 Sigmoid |

(a) MNIST dataset

| Encoder |
|---------|
| Input $\in \mathbb{R}^{3x32x32}$ |
| 4x4 conv. 32 stride 2 pad 1 & ReLU |
| 4x4 conv. 64 stride 2 pad 1 & ReLU |
| 4x4 conv. 128 stride 2 pad 1 & ReLU |
| 4x4 conv. L stride 1 pad 0, 4x4 conv. L stride 1 pad 0 |

| Decoder |
|---------|
| Input $\in \mathbb{R}^L$ |
| 4x4 upconv. 128 stride 1 pad 0 & ReLU |
| 4x4 upconv. 64 stride 2 pad 1 & ReLU |
| 4x4 upconv. 32 stride 2 pad 1 & ReLU |
| 4x4 upconv. 3 stride 2 pad 1 & Sigmoid |

(b) SVHN dataset.

| Encoder | Decoder |
|---------|---------|
| Input $\in \mathbb{R}^{2048}$ | Input $\in \mathbb{R}^L$ |
| FC. 1024 ELU | FC. 256 ELU |
| FC. 512 ELU | FC. 512 ELU |
| FC. 256 ELU | FC. 1024 ELU |
| FC. $L$, FC. $L$ | FC. 2048 |

(c) CUB image dataset.

| Encoder |
|---------|
| Input $\in \mathbb{R}^{1590}$ |
| Word Emb. 128 |
| 4x4 conv. 32 stride 2 pad 1 & BatchNorm2d & ReLU |
| 4x4 conv. 64 stride 2 pad 1 & BatchNorm2d & ReLU |
| 4x4 conv. 128 stride 2 pad 1 & BatchNorm2d & ReLU |
| 1x4 conv. 256 stride 1x2 pad 0x1 & BatchNorm2d & ReLU |
| 1x4 conv. 512 stride 1x2 pad 0x1 & BatchNorm2d & ReLU |
| 4x4 conv. L stride 1 pad 0, 4x4 conv. L stride 1 pad 0 |

| Decoder |
|---------|
| Input $\in \mathbb{R}^L$ |
| 4x4 upconv. 512 stride 1 pad 0 & ReLU |
| 1x4 upconv. 256 stride 1x2 pad 0x1 & BatchNorm2d & ReLU |
| 1x4 upconv. 128 stride 1x2 pad 0x1 & BatchNorm2d & ReLU |
| 4x4 upconv. 64 stride 2 pad 1 & BatchNorm2d & ReLU |
| 4x4 upconv. 32 stride 2 pad 1 & BatchNorm2d & ReLU |
| 4x4 upconv. 1 stride 2 pad 1 & ReLU |
| Word Emb.$^T$ 1590 |

(d) CUB-Language dataset.

Table 3: Unimodal encoder and decoder architectures.

**Encoder**

Input $\in \mathbb{R}^{3x32x64}$

4x4 conv. 32 stride 2 pad 1 & ReLU
4x4 conv. 64 stride 2 pad 1 & ReLU
4x4 conv. 128 stride 2 pad 1 & ReLU
1x4 conv. 128 stride 1x2 pad 0x1 & ReLU
4x4 conv. L stride 1 pad 0, 4x4 conv. L stride 1 pad 0

(a) MNIST-SVHN dataset.

**Encoder**

Input $\in \mathbb{R}^{1x32x192}$

4x4 conv. 32 stride 2 pad 1 & BatchNorm2d & ReLU
4x4 conv. 64 stride 2 pad 1 & BatchNorm2d & ReLU
4x4 conv. 128 stride 2 pad 1 & BatchNorm2d & ReLU
1x4 conv. 256 stride 1x2 pad 0x1 & BatchNorm2d & ReLU
1x4 conv. 512 stride 1x2 pad 0x1 & BatchNorm2d & ReLU
4x4 conv. L stride 1 pad 0, 4x6 conv. L stride 1 pad 0

(b) CUB Image-Caption dataset.

Table 4: Joint encoder architectures.

# G  Qualitative Results on MNIST-SVHN

## G.1  Generative Results & Marginal Likelihoods on MNIST-SVHN

### G.1.1  MMVAE

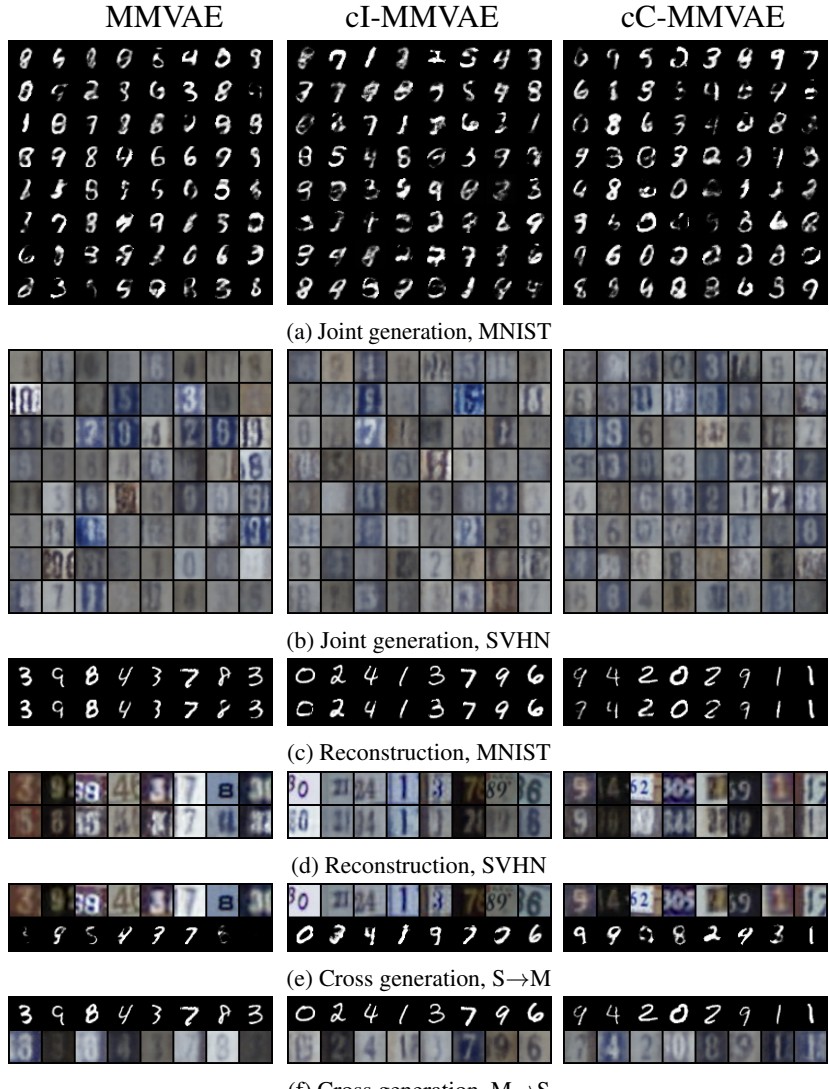

**MMVAE**  **cI-MMVAE**  **cC-MMVAE**

(a) Joint generation, MNIST

(b) Joint generation, SVHN

(c) Reconstruction, MNIST

(d) Reconstruction, SVHN

(e) Cross generation, S→M

(f) Cross generation, M→S

Figure 14: Generations of MMVAE model, from left to right are original model (MMVAE), contrastive loss with IWAE estimator (cI-MMVAE) and contrastive loss with CUBO estimator (cC-MMVAE).

| | | $\log p(\boldsymbol{x}_m, \boldsymbol{x}_n)$ | $\log p(\boldsymbol{x}_m \mid \boldsymbol{x}_m, \boldsymbol{x}_n)$ | $\log p(\boldsymbol{x}_m \mid \boldsymbol{x}_m)$ | $\log p(\boldsymbol{x}_m \mid \boldsymbol{x}_n)$ |
|---|---|---|---|---|---|
| $m$ = MNIST, $n$ = SVHN | MMVAE | $-1879.00$ | $-388.59$ | $-388.59$ | $-1618.53$ |
| | cI-MMVAE | $-1904.15$ | $-385.18$ | $-385.18$ | $-1620.77$ |
| | cC-MMVAE | $-1924.20$ | $-391.88$ | $-391.84$ | $-1619.34$ |
| $m$ = SVHN, $n$ = MNIST | MMVAE | $-1879.00$ | $-1472.44$ | $-1472.45$ | $-431.66$ |
| | cI-MMVAE | $-1904.15$ | $-1491.55$ | $-1491.56$ | $-444.28$ |
| | cC-MMVAE | $-1924.20$ | $-1490.56$ | $-1494.75$ | $-428.29$ |

Table 5: Evaluating log likelihoods using original model (MMVAE), contrastive loss with IWAE estimator (cI-MMVAE) and contrastive loss with CUBO estimator (cC-MMVAE). Likelihoods are estimated with IWAE estimator using 1000 samples.

### G.1.2 MVAE

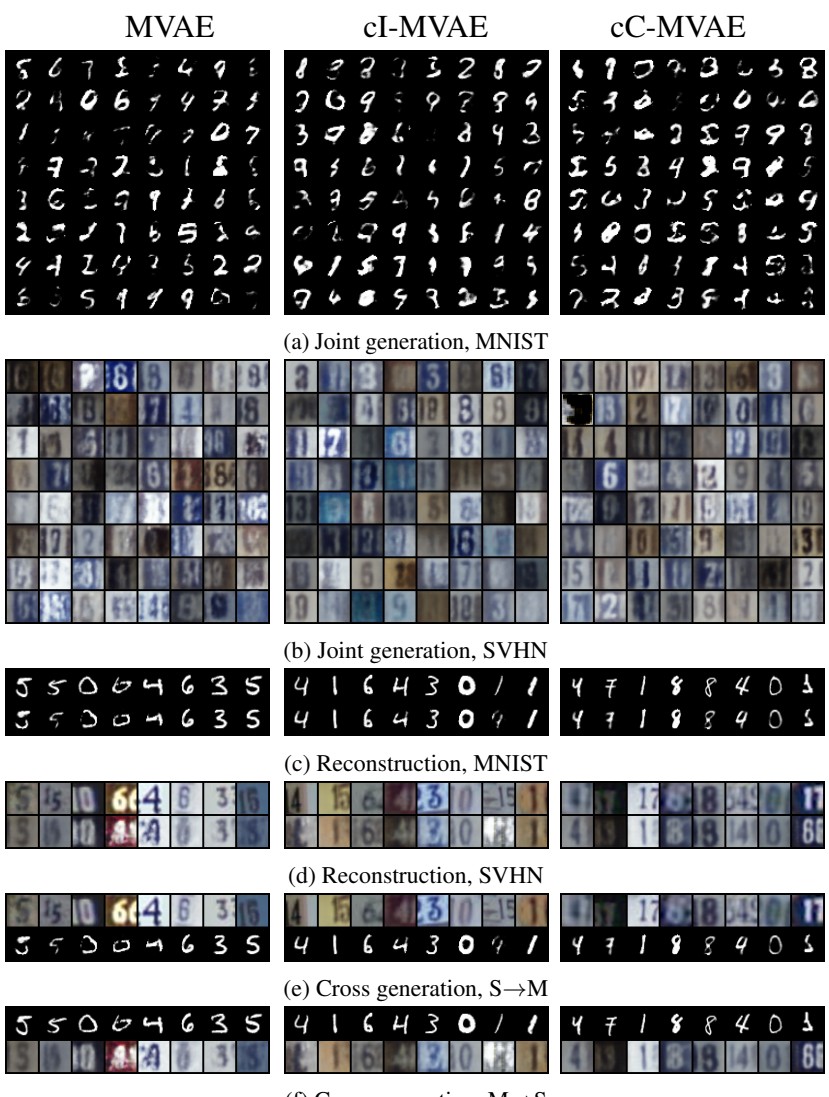

MVAE        cI-MVAE        cC-MVAE

(a) Joint generation, MNIST

(b) Joint generation, SVHN

(c) Reconstruction, MNIST

(d) Reconstruction, SVHN

(e) Cross generation, S→M

(f) Cross generation, M→S

Figure 15: Generations of MVAE model, from left to right are original model (MVAE), contrastive loss with IWAE estimator (cI-MVAE), contrastive loss with CUBO estimator (cC-MVAE).

| | | $\log p(\boldsymbol{x}_m, \boldsymbol{x}_n)$ | $\log p(\boldsymbol{x}_m \mid \boldsymbol{x}_m, \boldsymbol{x}_n)$ | $\log p(\boldsymbol{x}_m \mid \boldsymbol{x}_m)$ | $\log p(\boldsymbol{x}_m \mid \boldsymbol{x}_n)$ |
|---|---|---|---|---|---|
| $m = \text{MNIST}$, $n = \text{SVHN}$ | MVAE | $-404.43$ | $-404.43$ | $-388.27$ | $-1847.85$ |
| | cI-MVAE | $-406.85$ | $-406.22$ | $-388.26$ | $-1876.95$ |
| | cC-MVAE | $-405.24$ | $-432.96$ | $-388.26$ | $-1889.35$ |
| $m = \text{SVHN}$, $n = \text{MNIST}$ | MVAE | $-404.43$ | $-1518.00$ | $-1488.21$ | $-440.32$ |
| | cI-MVAE | $-406.85$ | $-1529.32$ | $-1498.73$ | $-443.04$ |
| | cC-MVAE | $-405.24$ | $-1520.23$ | $-1499.47$ | $-441.01$ |

Table 6: Evaluating log likelihoods using original model (MVAE), contrastive loss with IWAE estimator (cI-MVAE) and contrastive loss with CUBO estimator (cC-MVAE). Likelihoods are estimated with IWAE estimator using 1000 samples.

### G.1.3 JMVAE

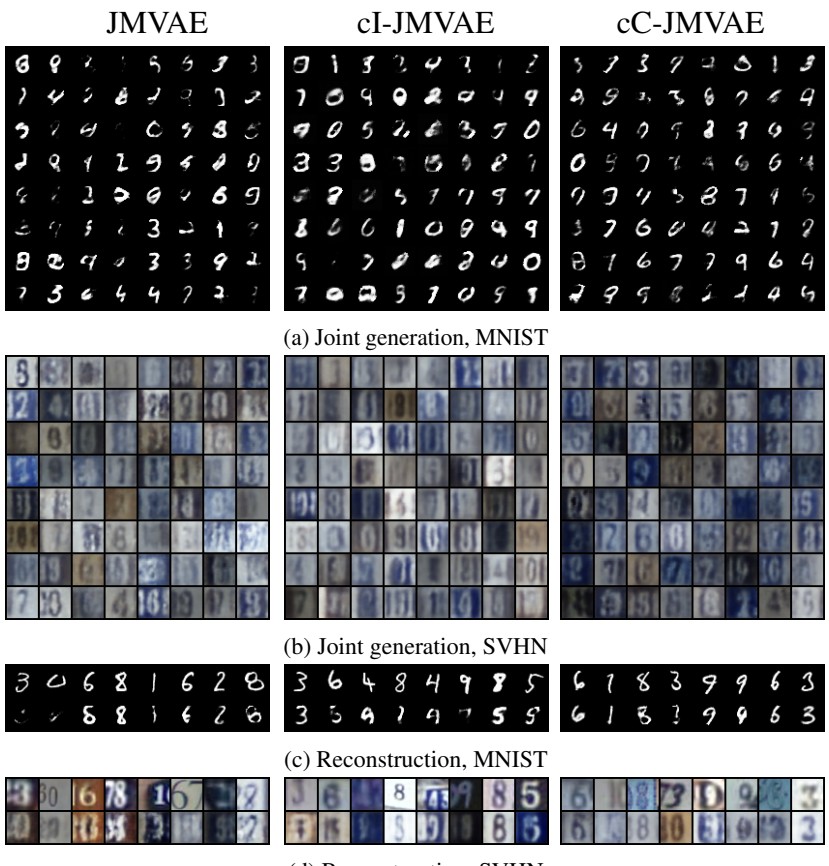

JMVAE      cI-JMVAE      cC-JMVAE

(a) Joint generation, MNIST

(b) Joint generation, SVHN

(c) Reconstruction, MNIST

(d) Reconstruction, SVHN

Figure 16: Generations of JMVAE model, from left to right are original model (JMVAE), contrastive loss with IWAE estimator (cI-JMVAE), contrastive loss with CUBO estimator (cC-JMVAE).

|  |  | $\log p(\boldsymbol{x}_m, \boldsymbol{x}_n)$ | $\log p(\boldsymbol{x}_m \mid \boldsymbol{x}_m, \boldsymbol{x}_n)$ |
|---|---|---|---|
| $m =$ MNIST, $n =$ SVHN | JMVAE | $-515.44$ | $-497.95$ |
|  | cI-JMVAE | $-518.56$ | $-511.28$ |
|  | cC-JMVAE | $-534.26$ | $-510.90$ |
| $m =$ SVHN, $n =$ MNIST | JMVAE | $-515.44$ | $-1515.73$ |
|  | cI-JMVAE | $-518.56$ | $-1529.44$ |
|  | cC-JMVAE | $-534.26$ | $-1614.78$ |

Table 7: Evaluating log likelihoods using original model (JMVAE), contrastive loss with IWAE estimator (cI-JMVAE) and contrastive loss with CUBO estimator (cC-JMVAE). Likelihoods are estimated with IWAE estimator using 1000 samples.

## G.2 GENERATION DIVERSITY

To further demonstrate that the improvements from our contrastive objective did not come at a price of sacrifising generation diversity, in Figure 17 we show histograms of the number of examples generated for each class with samples from prior. Similar to how we compute joint coherence, the class label of generated images are determined using classifiers trained on the original MNIST and SVHN dataset.

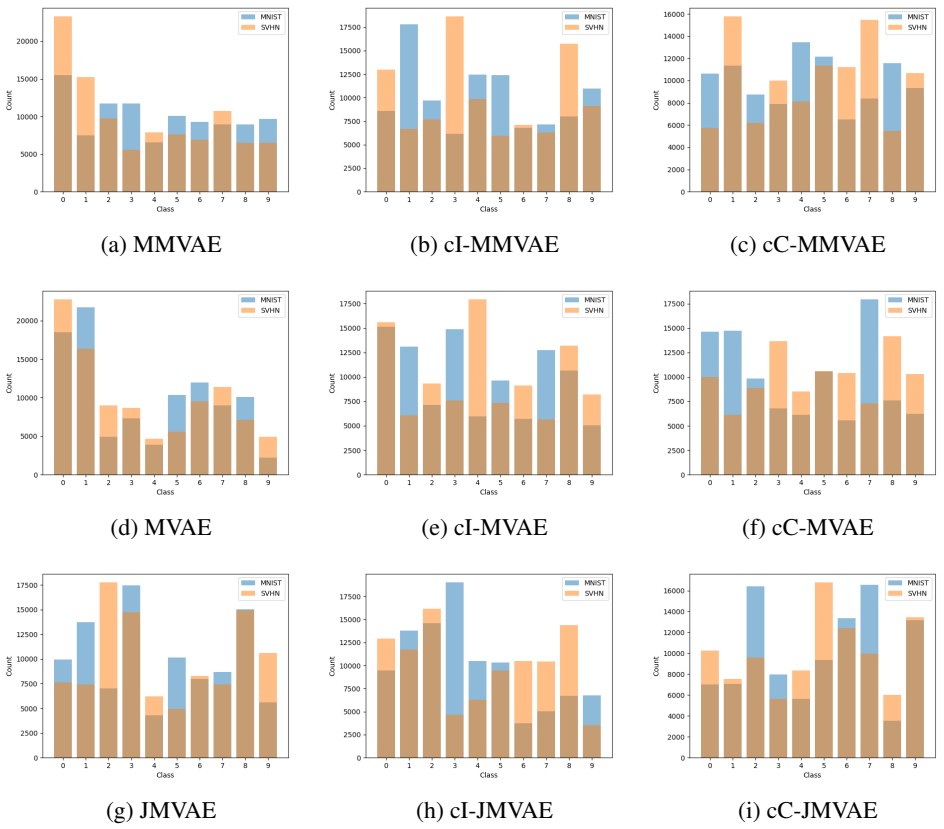

(a) MMVAE      (b) cI-MMVAE      (c) cC-MMVAE

(d) MVAE      (e) cI-MVAE      (f) cC-MVAE

(g) JMVAE      (h) cI-JMVAE      (i) cC-JMVAE

Figure 17: Number of examples generated for each of class during joint generation.

We can see that the contrastive models (cI-∗ and cI-∗) are capable of generating examples from different classes, and for MMVAE and MVAE the histogram of contrastive models are more uniform than the original models (less variance between class counts).

## H    QUALITATIVE RESULTS ON CUB

The generative results of MMVAE, cI-MMVAE and cC-MMVAE on CUB Image-Caption dataset are as shown in Figure 18, Figure 19 and Figure 20. Note that for the generation in the vision modality, we reconstruct and generate features from ResNet101 and perform nearest neighbour search in all the features in train set to showcase our generation results.

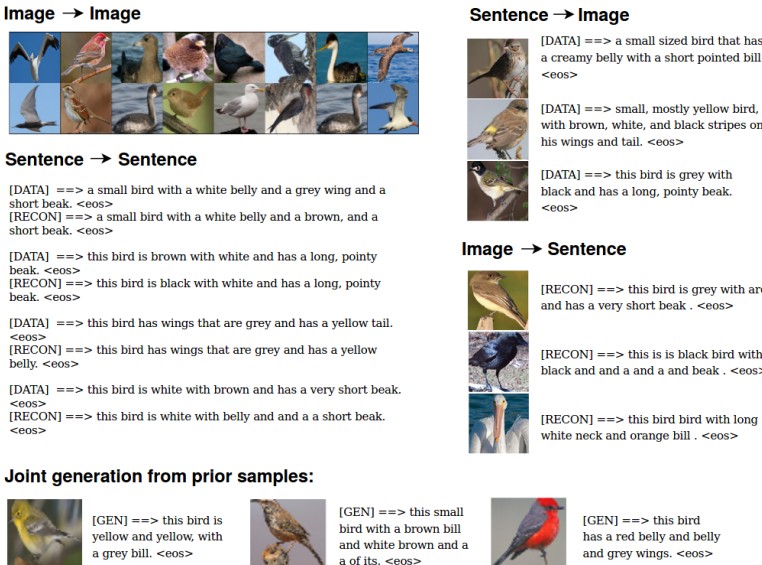

Figure 18: Qualitative results of MMVAE on CUB Image-Caption dataset, including reconstruction (vision → vision, language → language), cross generation (vision → language, language → vision) and joint generation from prior samples.

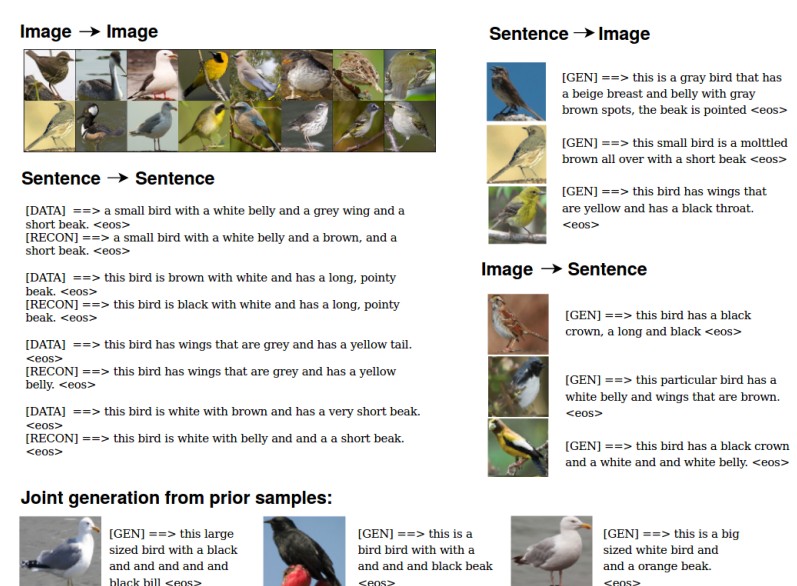

Figure 19: Qualitative results of MMVAE trained with contrastive loss with IWAE estimator on CUB Image-Caption dataset, including reconstruction (vision → vision, language → language), cross generation (vision → language, language → vision) and joint generation from prior samples.

**Image → Image**

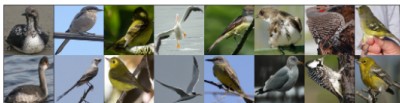

**Sentence → Sentence**

[DATA] ==> this bird has a belly that is white with brown sides . <eos>
[RECON] ==> this bird has a belly that is yellow and orange wings . <eos>

[DATA] ==> this bird has wings that are blue and has a white beak . <eos>
[RECON] ==> this bird has wings that are blue and has an orange bill . <eos>

[DATA] ==> the bird has a grey body and a white and grey speckled chest along with an orange beak . <eos>
[RECON] ==> the bird has a small bill , and black , and and grey and and and white feet . <eos>

[DATA] ==> a large and round bird with the colors of black and white feathers . <eos>
[RECON] ==> a white and white bird has a black brown black and white feet . <eos>

**Sentence → Image**

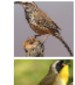 [DATA] ==> this gray speckled bird has very light gray legs, a black and white speckled breast and belly and a gray bill <eos>

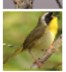 [DATA] ==> this bird has a belly that is whit with brown sides <eos>

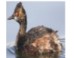 [DATA] ==> a large and round bird with the colors of black and white features <eos>

**Image → Sentence**

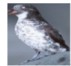 [RECON] ==> this bird is grey with are and has a very short beak . <eos>

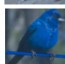 [RECON] ==> this is is black bird with a black and and a and a and beak . <eos>

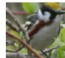 [RECON] ==> this bird bird with long white neck and orange bill . <eos>

**Joint generation from prior samples:**

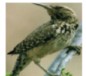 [GEN] ==> small yellow white grey and white bird with a short long beak beak.<eos>

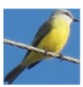 [GEN] ==> this bird is large yellow black with a yellow belly and <eos>

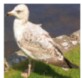 [GEN] ==> a bird with long bill bill, and and white, and brown crown. <eos>

Figure 20: Qualitative results of MMVAE trained with contrastive loss with CUBO estimator on CUB Image-Caption dataset, including reconstruction (vision → vision, language → language), cross generation (vision → language, language → vision) and joint generation from prior samples.

