# OpenReview forum: "Relating by Contrasting: A Data-efficient Framework for Multimodal Generative Models"
_ICLR.cc/2021/Conference — ICLR 2021 Poster_

### Official Review · AnonReviewer1 · 2020-10-27
**The paper deals with multimodal VAEs and addresses their problem of requiring lots of "related" (i.e., weakly-supervised) samples. To tackle the sample inefficiency, the paper proposes a contrastive objective.**

**Rating:** 6
**Confidence:** 4

**Review:**

 The paper proposes a contrastive objective that (1) minimizes the distance between "related" samples while (2) maximizing the distance between randomly paired samples. Existing multimodal VAEs optimize (1) via different multimodal ELBOs. The novelty lies in the optimization of (2) which can further benefit from unimodal samples for which no "related" samples of the other modality are available---this can be viewed as a semi-supervised approach for weakly-supervised multimodal data.  For the estimation of (2), the paper experiments with two different estimators, IWAE and CUBO.

The paper claims three contributions:
- (C1) improve multimodal learning of existing multimodal VAEs by extending them with objective (2)
- (C2) improve sample efficiency of multimodal VAEs by extending them with objective (2)
- (C3) further improve sample efficiency by bootstrapping "related" samples from a larger pool of unimodal samples

Overall, the empirical results support the claims of improved sample efficiency, but there might be a potential problem with claim (C1), due to the one-sided selection of metrics used for evaluation.


## Strong points

The paper tackles a practical problem (sample efficiency) with a
straightforward solution (leverage unpaired data) that can be easily
implemented and seems to work well across a family of models.

Strong empirical results demonstrating improved sample efficiency. For the
considered datasets, the proposed objective improves sample efficiency
significantly, requiring only 20% of the dataset to reach a
performance comparable to not using the additional objective. Additional
results showing that the sample efficiency can further be improved by a
procedure for bootstrapping paired samples from a set of unpaired samples.
This is an interesting idea that has been explored in semi-supervised learning,
but the present paper applies it to a weakly-supervised, multimodal data.

The paper nicely connects the objectives used in multimodal VAEs to contrastive
learning. It extends the different ELBOs used in multimodal VAEs with an
additional objective for minimizing the similarity of randomly paired samples.


## Weak points

My main objection is with regards to claim (C1), which states that the new
objective improves multimodal learning. The problem is that the choice of
metrics might be too one-sided, measuring only the "relatedness" in terms of
ground truth labels, but not the generative quality (of a generative model). As
such, the used metrics (linear classification accuracy; classification accuracy
of a pretrained classifier on generated samples) ignore the diversity of
generated samples. For example, a model could achieve perfect scores by
generating a single "canonical" image of the corresponding class, akin to mode
collapse. I understand that generative quality is not the paper's primary goal,
but it should not be overlooked, since otherwise, it begs the question of why to
use decoders at all and not just apply contrasting in latent space. The
ablations in the appendix indicate a trade-off between the "relatedness"
metrics and generative quality, suggesting that the proposed objective yields a
looser ELBO (Figure 13) and results in a worse generative performance (Figure
12). If such a trade-off exists, I would recommend presenting it more
transparently in the experiments and phrasing the respective claim more
conservatively.  Further, all previous multimodal VAEs report log-likelihood
values, so it seems natural to provide these numbers, too.

In the related work section, it is not clear how the authors conclude that contrastive methods are limited to "specific tasks" and how the proposed model overcomes this limitation. If the statement refers to the manual
design of a contrastive objective, I would argue that previous work includes
experiments showing that learned representations still generalize to other
downstream tasks, such as classification and object detection. If the authors
refer to generative tasks in particular, this should be stated more clearly.

Overall, it is a good paper with strong empirical results for the sample
efficiency argument. Yet, I tend towards a weak accept, due to the one-sided
evaluation. I will be happy to adjust my score if the paper presents the
generative results more transparently in the experiments.


## Questions

In the experiments of Section 4.3, do I understand correctly that when using
e.g. 20% of the data, you do not use any of the remaining 80% as "unrelated"
samples for contrasting? Put differently, I understand that you only use
randomly paired samples among the 20%-subset for that purpose, correct?

In the description of the datasets you mention that multiple pairings are used
to create a dataset of "related" samples.  Have you also tried to reduce the
number of pairings instead of taking $n$ percent of the dataset? Does this make
a difference with respect to classification, coherence, or generative
performance?

"Relatedness" in this work is based on log-likelihoods computed in sample
space.  However, contrasting in latent space might provide a more meaningful
measure of "relatedness". Have you considered computing the objective (or at
least the part that estimates the "unrelatedness") using latent representations
instead?

In Section 4.5 it is not quite clear how you estimate the "optimal threshold".
What measure is this threshold based on? Do the results in Figure 8 depend
on this threshold and would it be reasonable to show an ablation across
different threshold values?

In the paragraph above Equation (2) you mention the margin $m$.  It is not
clear why $m$ is not used later on. E.g., Is it an additional hyperparameter in
the model?


## Additional Feedback

In Hypothesis 3.1., you use the pointwise mutual information (PMI) and state
that PMI(x, y) > PMI(x, y'). As far as I understand, PMI is zero when two
outcomes are independent, but it can be negative when two outcomes are not
independent, which would violate the "uncontroversial assumption".
Alternatively, maybe one can use the mutual information between two random
variables instead of PMI between outcomes?

The related work section could benefit from a short paragraph on
weakly-supervised learning, which seems to be an important theme in the present
paper.

---

> ### Author Response · Authors · 2020-11-17
> **Response to Reviewer 1 (2/2)**
>
> > "Relatedness" in this work is based on log-likelihoods computed in sample space. However, contrasting in latent space might provide a more meaningful measure of "relatedness". Have you considered computing the objective (or at least the part that estimates the "unrelatedness") using latent representations instead?
>
> Yes, we have indeed considered doing this, however the latent contrastive approach has the following problems:
> 1. **Modelling perspective:** The motivation of applying contrastive loss on the latent space is to minimise the distance of latents from the same class of different modalities. However, it would be unreasonable to say that the entire latent code should be close to each other when the two samples are related, since latent of each modality contains style information private to each domain. One thing that would be reasonable try is to first explicitly split the latent into private and shared subspaces and put the contrastive constraint on the shared subspace only. We will investigate this in our future work;
> 1. **Empirical perspective:** If we add contrastive loss on the latent space as a regulariser for VAE learning, we must consider hyperparameters such as weighting and the choice of distance; one would imagine that the optimal hyperparameter will also change when the regulariser is applied to different multi-modal VAEs and different dataset, making it difficult to use.
>
>
> > In Section 4.5 it is not quite clear how you estimate the "optimal threshold". What measure is this threshold based on? Do the results in Figure 8 depend on this threshold and would it be reasonable to show an ablation across different threshold values?
>
> The threshold is computed with the PMI estimate, as stated at the bottom of page 7. This made intuitive sense to us since the motivation of the objective (as per hypothesis 3.1) is to maximise the difference of PMI between related and unrelated pair. We apologise for the confusion and will make this clearer in our updated draft.
>
> In addition to PMI threshold, we have also experimented with using joint marginal likelihood and it didn't work nearly as well -- we will include this result as an ablation study in the updated manuscript.
>
>
> > In the paragraph above Equation (2) you mention the margin. It is not clear why is not used later on. E.g., Is it an additional hyperparameter in the model?
>
> Apologies for the confusion! We omitted the margin m in our objective following common practice in this sort of max-margin scheme. We will clarify this in updated manuscript.
>
>
> > In Hypothesis 3.1., you use the pointwise mutual information (PMI) and state that PMI(x, y) > PMI(x, y'). As far as I understand, PMI is zero when two outcomes are independent, but it can be negative when two outcomes are not independent, which would violate the "uncontroversial assumption". Alternatively, maybe one can use the mutual information between two random variables instead of PMI between outcomes?
>
>
> We simply claim PMI(related pair) > PMI(unrelated pair). We intend this claim primarily as a measure of dependence, not correlation. This should not depend on the sign of the PMI, which can be anything when densities are involved.
>
> Theoretically speaking using mutual information here would have similar effect, however it requires taking the expectation under the model which is difficult.
>
> > The related work section could benefit from a short paragraph on weakly-supervised learning, which seems to be an important theme in the present paper.
>
> Indeed! Thank you for the suggestion, we will cite relevant weakly-supervised learning paper in our related work in our updated manuscript.
>
>
> > Overall, it is a good paper with strong empirical results for the sample efficiency argument. Yet, I tend towards a weak accept, due to the one-sided evaluation. I will be happy to adjust my score if the paper presents the generative results more transparently in the experiments.
>
> We certainly understand your concerns -- we'd like to re-iterate that the approach in this paper does not harm generative performance. We will include the following results to make our generative performance more transparent:
> - Generative results for MNIST-SVHN;
> - Marginal likelihood tables for generation;
> - Histogram of number of examples generated from samples from prior for each class to show diversity of joint generation.

---

> ### Author Response · Authors · 2020-11-17
> **Response to Reviewer 1 (1/2)**
>
> > My main objection is with regards to claim (C1), which states that the new objective improves multimodal learning. The problem is that the choice of metrics might be too one-sided, measuring only the "relatedness" in terms of ground truth labels, but not the generative quality (of a generative model). As such, the used metrics (linear classification accuracy; classification accuracy of a pretrained classifier on generated samples) ignore the diversity of generated samples.
>
> This is indeed a valid concern. We would like to clarify that our models trained using contrastive methods do not suffer from mode collapse, and the joint generation of the models do cover a wide diversity of examples. We will reassure the readers of this in the updated manuscript by adding/further highlighting the following components:
>
> - We will add qualitative results of the model's reconstructions and joint/cross generations (in fact, we had already included these as part of ablation studies for gamma on Page 15, but we will make these qualitative results a standalone section so that it can be easily found);
> - To showcase the diversity of the joint generation, we will include a histogram computing the percentage of each class generated from prior samples;
> - To ensure the readers that there is no mode collapse, we will include a table with figures of marginal likelihoods for both original and contrastive multi-modal VAEs.
>
> We would also like to point out the following: a model that performs well on "synergy" and "cross coherence" is very unlikely to suffer from mode collapse. This is because the classifier used to compute these scores are trained using *real data*, not *generated data* as the reviewer suggested. Therefore, if the model was indeed generating noise, it would be very unlikely that the classifier still matches the noise to the correct class.
>
> > The ablations in the appendix indicate a trade-off between the "relatedness" metrics and generative quality, suggesting that the proposed objective yields a looser ELBO (Figure 13) and results in a worse generative performance (Figure 12). If such a trade-off exists, I would recommend presenting it more transparently in the experiments and phrasing the respective claim more conservatively. Further, all previous multimodal VAEs report log-likelihood values, so it seems natural to provide these numbers, too.
>
> We will make the suggested changes in our updated manuscript to reflect that. Thank you for the suggestion.
>
>
> > In the related work section, it is not clear how the authors conclude that contrastive methods are limited to "specific tasks" and how the proposed model overcomes this limitation. If the statement refers to the manual design of a contrastive objective, I would argue that previous work includes experiments showing that learned representations still generalize to other downstream tasks, such as classification and object detection. If the authors refer to generative tasks in particular, this should be stated more clearly.
>
> By "specific tasks" we do indeed refer to a comparison to generative tasks and we will clarify this in the updated paper.
>
>
> > In the experiments of Section 4.3, do I understand correctly that when using e.g. 20% of the data, you do not use any of the remaining 80% as "unrelated" samples for contrasting?
>
> Yes, that is correct.
>
> > In the description of the datasets you mention that multiple pairings are used to create a dataset of "related" samples. Have you also tried to reduce the number of pairings instead of taking percent of the dataset? Does this make a difference with respect to classification, coherence, or generative performance?
>
> Yes, we have experimented with this. For two subsets of the original dataset that are of the same cardinality, the model performs better when the reduction of size is due to fewer pairings, compared to a lower percentage taken from each modality. This is because in the former case the model sees more variation in each modality.
>
> It is indeed interesting to consider this alternate way of reducing the dataset size as it casts light on an important question to be asked of the data --- whether it is easier to obtain relatedness through algorithmic pairing, or through manual annotation.
>
> If we consider this from a semi-supervised learning perspective (in MNIST-SVHN case), typically the fewer labels used for each modality the better, in which case the method used in our paper makes more practical sense; however if we consider the language-vision scenario where no explicit labels can be assigned for each example, then to reduce the size of the dataset one would have to reduce the number of pairings.

---

### Official Review · AnonReviewer4 · 2020-10-28
**concerns about the derivation and the connection between motivation and the final objective.**

**Rating:** 5
**Confidence:** 4

**Review:**

The paper tries to minimize the difference of the PMI between related and unrelated pairs of multimodal data but arrives at a very different objective with many approximations. It can be plugged into existing VAE based methods and improve learning performance and data efficiency. My major concern is about the derivation and the connection between motivation and the final objective.

Clarity and correctness:

The presentation of Section 3 should be further clarified. In particular, I wonder about the validness of Hypothesis 3.1. First, it is not clear to me which joint distribution is used in the definition of PMI in the hypothesis? It is not specified and I guess it is the model distribution after training according to the derivation in Appendix A. It is kind of strange because we make a hypothesis about the model after training and optimize the model to reach the hypothesis.

The derivation is also unclear, according to Eqn. (6), the objective should be the difference between both joint and marginal distributions. I'm not fully convinced by simply ignoring the difference between the marginal ones. I note that the authors mention that the marginal term is not relevant to the relatedness but ignoring it is not optimizing the difference between the PMI, right?

Eqn. (6) to Eqn. (2) is also unclear, why we optimize log sum p(negative) instead of optimizing sum log p(negative)? log sum negative is quite like an approximate of log marginal, which seems to contradict the discussion under Eqn. (6).

The resulting final objective (4) does not obviously connect to the original motivation: maximizing the difference between the PMI values of related and unrelated pairs. I checked Appendix B while the authors claim that it approximately optimizes the joint loglikelihood with an IPM regularization. I have several questions: i) how does this connect to the original motivation? ii) how does the approximation in Eqn. (7) affect training? iii) can we directly optimize Eqn. (7) instead of Eqn. (4)?

The estimate of the final objective also involves unknown approximation. In particular, the paper does not use CUBO and IWAE properly to form a valid bound of the objective.

With so many approximations, I can hardly figure out the key factors of the proposed method and say about the technical contribution.

Experiments:

I'm not an expert in multimodal learning. The paper mainly focuses on comparison with VAE based methods. The evaluation metrics are not commonly used but chosen following a related work. The paper claims that the main empirical contribution is its data efficiency, which seems to be verified.

===========

**after rebuttal**

Some of the issues are clarified. I updated my score to 5.

---

> ### Author Response · Authors · 2020-11-17
> **Response to Reviewer 4 (2/2)**
>
> > The resulting final objective (4) does not obviously connect to the original motivation: maximizing the difference between the PMI values of related and unrelated pairs. I checked Appendix B while the authors claim that it approximately optimizes the joint loglikelihood with an IPM regularization. I have several questions:
> > - how does this connect to the original motivation? how does the approximation in Eqn. (7) affect training? can we directly optimize Eqn. (7) instead of Eqn. (4)?
>
> What motivated our work is our interests in studying the effect of applying contrastive loss to generative models, and from the intuition that under the true data-generating distribution the PMI should be higher for related than for unrelated pairs. Section 3.1 describes our thought process which leads us from this starting point to the objective defined in Eqn. (4).
>
> This brings us to Appendix B. Appendix B is not a derivation of our objective, but rather an observation which should help us better understand it. As you point out, there are several approximations in section 3.1 --- we too would like to understand better the resulting objective in Eqn. (4), and do so by relating it to a maximum likelihood objective regularized by an approximation to the PMI. We never optimize Eqn. (7) directly, which is intractable due to the presence of the marginal likelihood terms; though, we agree that it could be interesting as future work to think about other approaches which approximate and optimize this directly.
>
> We will clarify this in a revision to the manuscript.
>
>
> > The estimate of the final objective also involves unknown approximation. In particular, the paper does not use CUBO and IWAE properly to form a valid bound of the objective.
>
> This is not true. We went to significant lengths to formulate and evaluate a valid bound to the contrastive objective. You can find details about this in Section 3.2 and all of our experiments.
>
> In particular, we describe in item 2 of Section 3.2 that by employing the CUBO upper bound estimator for term 2 of Eqn. 4, the approximation **is** a proper upper-bound of the objective.
> We also formulate an alternative, using the IWAE estimator for both terms of Eqn. 4 (item 1 of Section 3.2). While this means we no longer have a bounded approximation, it has the desirable quality of being an arbitrarily tight and low-variance estimator of the marginal likelihood.
> In our empirical evaluation we showcased the pros and cons of using these two different approximations with the `cI-*` (IWAE) and `cC-` (CUBO) models.
>
> Note that it is common practice to approximate the intractable marginal likelihood in VAEs, and our proposal of having a contrastive objective doesn't change that.
>
>
> > With so many approximations, I can hardly figure out the key factors of the proposed method and say about the technical contribution.
>
> While we sympathise with the reviewer's comments about approximations, the only place where approximation appears in the derivation of our objective is the marginal likelihood approximation described in Section 3.2, which is a property of the VAE, not of our method. Other approximations only appear in Appendix A and B, which are intended to provide intuitions for our proposed objective.
>
> Our technical contribution is the novel application of contrastive framework to VAE objective with careful consideration of optimisation, which we believe, with the positive empirical evaluations, has been demonstrated to be valid and effective.

---

> > ### Comment · AnonReviewer4 · 2020-11-20
> > **Thanks for your feedback. Additional things to be clarified.**
> >
> > First, thanks to the authors for their detailed feedback. I have read the revised version and rebuttal.
> >
> > Some of the issues are clarified. Herewith some additional feedback.
> >
> > #### Q1.
> >
> > The authors try to clarify that **the final objective makes sense because it characterizes the relatedness among/between modalities**.  This is one of the central issues of the paper in my opinion. There are two perspectives provided in the paper while neither is "perfect".
> >
> > The first one is about the derivation in Appendix A. However, both the authors and I agree that *relatedness is not equivalent to PMI* because of ignoring something about the marginal in a single modality.  Simply ignoring such terms means that the authors consider the "relatedness" but the real mechanism is not clear. Or at least not as clear as PMI is.
> >
> > The second one is about the derivation in Appendix B. The authors and I agree that the approximation is not clear either.
> >
> > #### Q2.
> >
> > I realize it is a misunderstanding of the CUBO and IWAE things in Sec. 3.2. Thanks for clarifying this. Currently, according to the rebuttal, I believe that the cC method is a valid bound of the original objective and cI is a "wrong" but may be of lower variance estimate. However, the presentation of the revised paper is still misleading in my opinion. I suggest the authors to further clarify this.
> >
> > #### Q3.
> >
> > Another issue is that whether Hypothesis 3.1can be validated in some sense (empirically / theoretically)? Or in which scope it holds or not?

---

> > > ### Author Response · Authors · 2020-11-21
> > > **Response to additional questions**
> > >
> > > We thank the reviewer for additional feedback! Below are our response to the three questions raised:
> > >
> > > **Q1.**
> > > As we stated, what motivated our work is our interests in studying the effect of applying contrastive loss to generative models. Therefore, conceptually, we consider relatedness similar to how it is considered in the contrastive loss literature: that the two inputs should have conceptual commonality. More formally, we characterised relatedness with the difference between pointwise mutual information (PMI) of related vs. unrelated pairs in Hypothesis 3.1. As for the "real mechanism" of relatedness, it should defer in different multimodal scenarios: for MNIST-SVHN it is digits, for language-vision dataset it is the object of interest.
> > >
> > > Appendix A and B are not how we characterise relatedness, but rather, provide intuition to our final objective.
> > >
> > > **Q2.**
> > > We thank the reviewer for re-investigating Sec 3.2 and we are glad that we could clarify the misunderstanding. We will update the manuscript to highlight the fact that we have 2 versions of objective: one a valid bound with high variance and the other a non-bound with arbitrarily low variance.
> > >
> > > **Q3.**
> > > Hypothesis 3.1 can indeed be validated. In fact, the label propagation experiments in Sec 4.5 provide quite strong empirical evidence to our claim. We find for a generative model trained with contrastive objective, the difference between PMI for related pairs and unrelated pairs is a good indicator of relatedness, enabling label propagation by thresholding the PMI estimated by the trained model. This shows that Hypothesis 3.1 is true for a well-trained parametrised generative model.
> > >
> > > We thank the reviewer for enquiring about this connection; we will update our manuscript to highlight this and make our motivations clearer.

---

> ### Author Response · Authors · 2020-11-17
> **Response to Reviewer 4 (1/2)**
>
> > I wonder about the validness of Hypothesis 3.1. First, it is not clear to me which joint distribution is used in the definition of PMI in the hypothesis? It is not specified and I guess it is the model distribution after training according to the derivation in Appendix A. It is kind of strange because we make a hypothesis about the model after training and optimize the model to reach the hypothesis.
>
> Hypothesis 3.1 is defined for the true generative model, and our goal is to learn a parametric model that targets this true generative process by setting up the model for multimodal data and learning such a model with a concomitant objective.
> We will update the manuscript to ensure that this is made clear; thank you for pointing this out!
>
> > The derivation is also unclear, according to Eqn. (6), the objective should be the difference between both joint and marginal distributions. I'm not fully convinced by simply ignoring the difference between the marginal ones. I note that the authors mention that the marginal term is not relevant to the relatedness but ignoring it is not optimizing the difference between the PMI, right?
>
> Yes, you are correct in saying that we are not optimising the difference between PMI. As we stated in our paper, we are only optimising the components in PMI difference that are relevant to relatedness (or, to say the least, involves two different modalities), since learning relatedness is our goal. The $log p(y') - log p(y)$ term we discarded in PMI difference (term 2 of Eqn. 6) involves only one modality (functionally, it captures the typicality within modality) and is therefore not relevant to our goal of learning relatedness.
>
>
> > Eqn. (6) to Eqn. (2) is also unclear, why we optimize log sum p(negative) instead of optimizing sum log p(negative)? log sum negative is quite like an approximate of log marginal, which seems to contradict the discussion under Eqn. (6).
>
> We justify the choice of the log sum instead of sum log form of objective with the following three arguments:
> Note that Eqn. 2 can be rewritten as $log p(x,y)/\sum_y' p(x,y')$, which can be seen as the logarithm of the ratio between the positive term and the sum of negative terms. This form of contrastive loss is commonly seen in literature of self-supervised contrastive learning (e.g. [1-3]); a consequence of `LogSumExp` being a smooth approximation to the `max` function [4].
>
> Moreover, from Jensen's inequality, the proposed approach (sum outside the log) is upper bounded by our formulation (`LogSumExp`). We chose our formulation simply as a matter of convention.
>
> [1] Representation Learning with Contrastive Predictive Coding, https://arxiv.org/abs/1807.03748
>
> [2] Learning deep representations by mutual information estimation and maximization, https://arxiv.org/abs/1808.06670
>
> [3] A Simple Framework for Contrastive Learning of Visual Representations, https://arxiv.org/abs/2002.05709
>
> [4] A Metric Learning Reality Check, https://arxiv.org/abs/2003.08505

---

### Official Review · AnonReviewer3 · 2020-10-28
**Multimodal VAE using contrastive-style objective.**

**Rating:** 7
**Confidence:** 4

**Review:**

This work presents a generative model for multimodal learning. The paper maximizes or minimizes the pointwise mutual information between data from two modalities considering a novel random variable relatedness to dictates if data are related or not.  This is realized by casting multimodal learning as max-margin optimization with the contrastive loss for the objective. For the optimization, the paper considers the IWAE estimator. As per the experiments, the paper considers MNIST-SVHN and CUB Image-Captions dataset and perform evaluations across four metrics. Using the experiments, the paper demonstrates that the proposed approach improves multimodal learning, data-efficient learning, and label propagation.

- The paper is well written and easy to follow.
- The experimental design is praiseworthy as the paper considers different aspects in evaluating multimodal learning via a generative model. The optimization of the proposed approach is also carefully considered.
- The experimental results across all three tasks demonstrate the efficacy of the proposed approach.
- One weakness that I find is a lack of comparison against the GAN based models. Can the authors do such a comparison? Or at least discuss why and how GAN based models could be integrated with the current approach?

---

> ### Author Response · Authors · 2020-11-17
> **Response to Reviewer 3**
>
> > One weakness that I find is a lack of comparison against the GAN based models. Can the authors do such a comparison? Or at least discuss why and how GAN based models could be integrated with the current approach?
>
> Thank you for the suggestion and positive appraisal.
>
> We didn't cover GANs as we were primarily interested in the representation-learning aspects of the multimodal VAE rather than the quality of generations of the model. One could potentially extend this idea to GANs by employing an additional discriminator that evaluates relatedness between generations across modalities, but this is beyond the scope of our work here. We can however include a discussion of GANs and their applicability in this context to the updated manuscript.

---

### Official Review · AnonReviewer2 · 2020-10-29
**Interesting contribution on Multimodal VAEs**

**Rating:** 6
**Confidence:** 4

**Review:**

Summary

The paper proposes a contrastive multimodal generative model framework for including unrelated datapoints as well as related datapoints in the learning of the multimodal model. Using such a contrastive formulation, the paper shows that the proposed model outperforms previous work on four desiderata for multimodal generative models.

Strengths
+ The paper is a novel take on contrastive learning in a probabilistic generative modeling framework, which is interesting
+ The paper reports extensive experimental results

Weaknesses

Why not directly use some energy based model with a contrastive loss to learn such multimodal generative models? It feels somewhat odd to both normalize p(X, Y) and also require unaligned examples as supervision. With an energy based model one can do joint generation, conditional generation (coherence) and synergy studies. A discussion on this would be really helpful!

A minor point, one could also consider the case where “r” is latent, that would actually bring up a very relevant case in vision and language where one crawls the web and the vision and language pairs may be apriori related or unrelated (but it is not known if they are or not). Also, intuitively, why is it that using unrelated data makes us believe that the models should be more data efficient? It would be great to give more intuition on that point.

Eqn. 2: The way the equation is written down (given the motivation above) it is not clear that the sum in the second term should be inside the log. Why was this choice made, instead of a more standard contrastive learning approach of summing over the distance with the negative examples (which would make the sum outside the log?). Also, paragraph 1 page 4 is very confusing as the model is talked about as generating when training but as I understand there is no generation when training (only computation of likelihoods?).

Appendix B proof: It appears that the proof might be incorrect as the sum over X that we have in Eqn. 4 is not over all X but over a sample of N examples randomly chosen. It then appears incorrect to claim that this is the same as PMI (as opposed to some kind of an approximation?).

Baseline explanation: It is not clear how the proposed approach is applied to the JMVAE objective (Table. 1), which adds additional terms to the regular ELBO. Are those additional terms also used? Clarification on that would be very helpful.

Overall, it would be very useful to get more insights on why the proposed approach is expected to yield a better multimodal generative model based on the four criteria. Is the idea that generative models without explicitly being given negative supervision are optimistic in which datapoints they consider to be related? How does contrastive training affect "coverage" metric as discussed in Vedantam et.al. (Triple ELBO)? Essentially, any insight on when one should and should not do contrastive training would be useful to add to the paper.

**POST REBUTTAL**
I read through the other reviews and the author rebuttal. I thank the authors for addressing all of my concerns in the rebuttal. Overall, this is an interesting contribution in the multimodal VAE space and would make for a good poster at the conference, and I am happy to keep my original rating for the paper.

In terms of energy based models here is one recent work which comes to mind which might be useful to extend to multimodal settings:
Du, Yilun, Shuang Li, and Igor Mordatch. 2020. “Compositional Visual Generation and Inference with Energy Based Models.” arXiv [cs.CV], April. https://arxiv.org/abs/2004.06030.
(note the quality of generations in Fig. 5)

---

> ### Author Response · Authors · 2020-11-17
> **Response to Reviewer 2 (2/2)**
>
> > It would be very useful to get more insights on why the proposed approach is expected to yield a better multimodal generative model based on the four criteria. Is the idea that generative models without explicitly being given negative supervision are optimistic in which datapoints they consider to be related?
>
>
> In a manner of speaking, yes.
>
> The four evaluation criteria we adopt from Shi et al. 2019 characterise the extent and effectiveness with which the latent variable captures information from the observed modalities and disentangles shared and modality-specific information.
>
> Non-contrastive approaches (e.g. Shi et al. 2019, Wu & Goodman 2018) only ever observe positive examples of relatedness, and are thus limited in their ability to implicitly capture what things ought not to be related. Contrastive methods, following the max-margin style of objective, can leverage unrelated data to better capture and disentangle representations, leading to better models as evaluated by the criteria, as observed in our experimental results.
>
>
> > How does contrastive training affect "coverage" metric as discussed in Vedantam et.al. (Triple ELBO)?
>
> The multimodal scenario considered in Vedantam et al. is a visual-attribute one: one modality is typically visual, such as human faces, and the other is a one hot attribute vector (e.g. gender or hair colour). Since these attribute labels are well-defined and categorical, it is easy to evaluate "coverage" . However, in the case of more generic multimodal data as considered here, it is unclear what would constitute "coverage". This was one of the primary reasons we evaluated on the four metrics (section 4.2) originally proposed in Shi et al. (2019).
>
>
> > Any insight on when one should and should not do contrastive training would be useful to add to the paper.
>
> As we show in our experimental results (Table 1 & 2, Figure 6), contrastive approach performs better than original models no matter the percentage of data used, indicating that it is always beneficial to adopt such a learning scheme even when more related data is available. However, there are two factors that broadly dictate if one can/ought to employ a contrastive approach:
>
> 1. Complexity of data and number of modalities:
>
>    While it is relatively straightforward to extend the 2-modality case discussed here by
>    considering the pointwise total correlation (PTC) instead of PMI, and evaluating the
>    appropriate number of marginals for the negative terms of the contrastive loss, such
>    approaches can quickly run into practical difficulties with the amount of computation
>    required. A similar argument can apply for more complex data like videos.
>
>    In saying this, we would also like to note that when the model is chosen sensibly it is
>    entirely possible to keep the computational cost quite low --- the most complicated
>    experiment ran on MMVAE can be trained on a 4GB GPU within a few hours.
>
> 2. Pervasiveness of 'relatedness'
>
>    Constructing the contrastive loss relies on the assumption that randomly sampled
>    pairs/tuples are typically unrelated. Were this not the case, the max-margin approach
>    can lead to poor representations.
>
> If both factors can be circumvented, using the contrastive objective is preferable.

---

> ### Author Response · Authors · 2020-11-17
> **Response to Reviewer 2 (1/2)**
>
> > Why not directly use some energy based model with a contrastive loss to learn such multimodal generative models? It feels somewhat odd to both normalize p(X, Y) and also require unaligned examples as supervision. With an energy based model one can do joint generation, conditional generation (coherence) and synergy studies. A discussion on this would be really helpful!
>
> While it is indeed true that contrastive methods have been typically used to train energy-based models, typically Restricted Boltzmann Machines (RBMs), such models have typically not seen much application in complex domains such as image/language. Was there perhaps some particular model you had in mind that would apply in this instance? We'd be happy to explore such a model.
>
> > One could also consider the case where “r” is latent, ...
>
> Indeed. Our primary interest here was the effect of adapting a contrastive objective for generative models like VAEs. We discovered that doing so is functionally similar to capturing an _implicit_ relatedness latent variable whose effect manifests through maximising (related) or minimising (unrelated) the likelihood; an observation we subsequently leverage for experiments that bootstrap off of available labelled related data.
> An explicit relatedness latent is an interesting direction to explore and we will be studying this in our future work. Thank you for the suggestion!
>
> > Eqn. 2: The way the equation is written down (given the motivation above) it is not clear that the sum in the second term should be inside the log. Why was this choice made, instead of a more standard contrastive learning approach of summing over the distance with the negative examples (which would make the sum outside the log?).
>
>
> Note that Eqn. 2 can be rewritten as $log p(x,y)/\sum_{y'} p(x,y')$, which can be seen as the logarithm of the ratio between the positive term and the sum of negative terms. This form of contrastive loss is commonly seen in literature of self-supervised contrastive learning (e.g. [1-3]); a consequence of `LogSumExp` being a smooth approximation to the `max` function [4].
>
> Moreover, from Jensen's inequality, the proposed approach (sum outside the log) is upper bounded by our formulation (`LogSumExp`). We chose our formulation simply as a matter of convention.
>
> [1] Representation Learning with Contrastive Predictive Coding, https://arxiv.org/abs/1807.03748
>
> [2] Learning deep representations by mutual information estimation and maximization, https://arxiv.org/abs/1808.06670
>
> [3] A Simple Framework for Contrastive Learning of Visual Representations, https://arxiv.org/abs/2002.05709
>
> [4] A Metric Learning Reality Check, https://arxiv.org/abs/2003.08505
>
>
> > Also, paragraph 1 page 4 is very confusing as the model is talked about as generating when training but as I understand there is no generation when training (only computation of likelihoods?).
>
> Indeed, there are no generations in the examples. We intend Figure 4 to highlight the fact that an image need not be meaningful for the likelihood to be minimised, and thereby provide an intuition on why the minimising log likelihood term in our objective can be overpowering.
>
> We will revise the narrative and clarify this aspect in a better manner.
>
>
> > Appendix B proof: It appears that the proof might be incorrect as the sum over X that we have in Eqn. 4 is not over all X but over a sample of N examples randomly chosen. It then appears incorrect to claim that this is the same as PMI (as opposed to some kind of an approximation?).
>
>
> Appendix B is intended to provide a more intuitive understanding of the objective in Eqn (4), through an alternate interpretation. It is indeed intended as an approximation as denoted in Eqn (7). We will make this clearer in the updated manuscript.
>
> > Baseline explanation: It is not clear how the proposed approach is applied to the JMVAE objective (Table. 1), which adds additional terms to the regular ELBO. Are those additional terms also used?
>
> Yes, the auxiliary terms of JMVAE are also used and we only replaced the ELBO term by our contrastive ELBO in Eqn 4.

---

### Author Response · Authors · 2020-11-19
**Summary of revisions based on rebuttal feedback**

We would like to thank the reviewers for their helpful suggestions. Based on the feedback we received, we have updated our manuscript with the following changes:
- Added qualitative results for our experiments in Appendix G and H, including:
    - Generative results (reconstruction, cross generation and joint generation);  **(R1)**
    - Marginal likelihood tables evaluated with IWAE (K=1000); **(R1)**
    - Sample per class histogram of joint generation to showcase generation diversity.  **(R1)**
- Added additional related work discussing weakly supervised methods; **(R1)**
- - Clarified in related work that previously seen contrastive approaches are not just good at "specific tasks", but also perform well at generalising to different downstream tasks; **(R1)**
- Added reason and relevant citations for choosing $\log \sum$ over $\sum \log$ for our objective in Eqn 2; **(R2, R4)**
- Clarified that Figure 4 only contains natural images; **(R2)**
- Included discussion on possible adaptation of our method for GANs in Section 5; **(R3)**
- Updated Appendix B to clarify that the PMI term in Eqn. 7 is an approximation; **(R2, R4)**
- Clarified the purpose of Hypothesis 3.1 --- the hypothesis should hold true for *true generative models*, and our goal is to learn a parametric generative model that matches this behaviour; **(R4)**
- Clarified the omitting of margin `m` in methodology. **(R1)**

We hope that these updates improves the clarify of our paper.

---

> ### Author Response · Authors · 2020-11-23
> **Summary of additional revisions based on R4 feedback**
>
> We updated the manuscript again based on Reviewer 4's additional feedback with the following changes:
>
> 1. Further clarification on the two estimators considered for term 2 of Eqn. 4--- one a valid bound with high variance (CUBO) the other a non-bound with arbitrarily low variance (IWAE) at the end of sec 3.2;
> 2. Highlight connection between Hypothesis 3.1 and the method used in label propagation experiments to provide empirical support to our hypothesis.
>
> We thank the reviewer again for the helpful feedback!

---

### Decision · Program_Chairs · 2021-01-07
**Final Decision**

**Decision:**

Accept (Poster)

**Comment:**

The paper proposes an efficient method to train generative models on multimodal data using a contrastive approach. Usually training such models requires significant training data to be able to learn patterns. The authors propose a variational autoencoder approach that enables multimodal learning of models using a data-efficient approach, and shows the effectiveness of the approach on challenging datasets.

The authors have mostly addressed the feedback of the reviewers and done some of the necessary changes to the paper (e.g., adding more results and missing related work). They should make sure to address any lingering concerns about the paper, mentioned by the reviewers in their post-rebuttal feedback.